# Involvement of Ca$_V$2.2 channels and $\alpha_2\delta$-1 in homeostatic synaptic plasticity in cultured hippocampal neurons

Kjara S. Pilch, Krishma H. Ramgoolam and Annette C. Dolphin

*Department of Neuroscience, Physiology and Pharmacology, University College London, London, UK*

Handling Editors: Katalin Toth & Samuel Young

The peer review history is available in the Supporting information section of this article (https://doi.org/10.1113/JP283600#support-information-section).

**Kjara S. Pilch** obtained her MSc in Neuroscience at the University of Bordeaux where she studied the role of purinergic receptors in synaptic plasticity and disease. She then joined the Dolphin lab at the Department of Neuroscience, Physiology and Pharmacology at University College London for her PhD. Here, she continued working on synaptic plasticity, focusing on the role of presynaptic voltage-gated Ca$_V$2.2 channels and their auxiliary subunit $\alpha_2\delta$-1 in homeostatic synaptic plasticity.

K. S. Pilch is eligible for Early Investigator Price.

This article was first published as a preprint. Pilch KS, Ramgoolam KH, Dolphin AC. 2022. Involvement of Ca$_V$2.2 channels and $\alpha_2\delta$-1 in hippocampal homeostatic synaptic plasticity. bioRxiv. https://doi.org/10.1101/2022.06.27.497782.

The Journal of Physiology

**Abstract**  In the mammalian brain, presynaptic $Ca_V2$ channels play a pivotal role in synaptic transmission by mediating fast neurotransmitter exocytosis via influx of $Ca^{2+}$ into the active zone of presynaptic terminals. However, the distribution and modulation of $Ca_V2.2$ channels at plastic hippocampal synapses remains to be elucidated. Here, we assess $Ca_V2.2$ channels during homeostatic synaptic plasticity, a compensatory form of homeostatic control preventing excessive or insufficient neuronal activity during which extensive active zone remodelling has been described. We show that chronic silencing of neuronal activity in mature hippocampal cultures resulted in elevated presynaptic $Ca^{2+}$ transients, mediated by increased levels of $Ca_V2.2$ channels at the presynaptic site. This work focused further on the role of $\alpha_2\delta$-1 subunits, important regulators of synaptic transmission and $Ca_V2.2$ channel abundance at the presynaptic membrane. We found that $\alpha_2\delta$-1 overexpression reduces the contribution of $Ca_V2.2$ channels to total $Ca^{2+}$ flux without altering the amplitude of the $Ca^{2+}$ transients. Levels of endogenous $\alpha_2\delta$-1 decreased during homeostatic synaptic plasticity, whereas the overexpression of $\alpha_2\delta$-1 prevented homeostatic synaptic plasticity in hippocampal neurons. Together, this study reveals a key role for $Ca_V2.2$ channels and novel roles for $\alpha_2\delta$-1 during synaptic plastic adaptation.

(Received 15 July 2022; accepted after revision 8 November 2022; first published online 14 November 2022)

**Corresponding authors** K. S. Pilch and A. C. Dolphin: Andrew Huxley Building, University College London, Gower Street, London WC1E 6BT, UK.    Emails: kjara.pilch.18@ucl.ac.uk and a.dolphin@ucl.ac.uk

**Abstract figure legend** $Ca_V2.2$ channels (green) are voltage-gated calcium channels that mediate the influx of $Ca^{2+}$ (red arrows) into the presynaptic terminal, which leads to neurotransmitter release. Their auxiliary $\alpha_2\delta$-1 subunits (purple) are crucial for the positioning and function of $Ca_V2.2$ channels (top panel). During homeostatic synaptic plasticity after chronic activity blockade, adaptational changes at hippocampal presynapses in culture include increases in both $Ca^{2+}$ flux and $Ca_V2.2$ channel abundance and a decrease in the $\alpha_2\delta$-1 subunits (bottom panel). These results provide novel insights into presynaptic function and changes at presynaptic terminals during synaptic plasticity.

## Key points

- The roles of $Ca_V2.2$ channels and $\alpha_2\delta$-1 in homeostatic synaptic plasticity in hippocampal neurons in culture were examined.
- Chronic silencing of neuronal activity resulted in elevated presynaptic $Ca^{2+}$ transients, mediated by increased levels of $Ca_V2.2$ channels at presynaptic sites.
- The level of endogenous $\alpha_2\delta$-1 decreased during homeostatic synaptic plasticity, whereas overexpression of $\alpha_2\delta$-1 prevented homeostatic synaptic plasticity in hippocampal neurons.
- Together, this study reveals a key role for $Ca_V2.2$ channels and novel roles for $\alpha_2\delta$-1 during synaptic plastic adaptation.

## Introduction

Synaptic communication relies on the translation of electrical signals to neurotransmitter release (Südhof & Rizo, 2011). For this, $Ca^{2+}$ enters the presynaptic active zone via voltage-gated $Ca^{2+}$ ($Ca_V$) channels, ultimately resulting in the release of synaptic vesicles. This process is tightly regulated by a multitude of proteins but must also remain dynamic to allow rapid adaptation to changes in signalling, such as synaptic potentiation or depression. As excessive or inadequate neuronal firing could impair neuronal activity and brain function, homeostatic synaptic plasticity (HSP) processes maintain

firing rates in a physiological range (Turrigiano, 2008). HSP mechanisms involve changes at both pre- and post-synaptic sites, such as direct regulation of synaptic inputs (Fernandes & Carvalho, 2016) and alteration of intrinsic neuronal excitability (Turrigiano, 2011), for example via presynaptic $Ca_V$ channels.

In the mammalian brain, both $Ca_V2.1$ (P/Q-type) and $Ca_V2.2$ (N-type) channels, each with distinct biophysical properties, provide the main sources of $Ca^{2+}$ influx at the presynapse (Dolphin & Lee, 2020). The distribution of each subtype varies depending on age, brain region, presynaptic action potential (AP) duration and synaptic activity, with some presynaptic terminals exclusively

expressing $Ca_V2.1$ or $Ca_V2.2$ (Bean, 2007; Dolphin & Lee, 2020). Most synaptic transmission, however, is likely mediated by the joint activity of $Ca_V2.1$ and $Ca_V2.2$ channels, enabling a diversification and fine-tuning of synaptic signalling at central synapses (Nakamura et al., 2015). Due to the relationship between the number of $Ca_V2$ channels at the active zone and vesicular release, $Ca_V2$ channels play a central role in homeostatic synaptic adaptations.

To study HSP, the sodium channel inhibitor tetrodotoxin (TTX) is widely used to induce chronic silencing of neuronal networks *in vitro* (Turrigiano, 2011). The prolonged application of TTX has been shown to increase presynaptic $Ca^{2+}$ flux (Zhao et al., 2011) and release probability (Vitureira et al., 2011), and induce restructuring of the active zone, involving multiple presynaptic proteins (Glebov et al., 2017; Lazarevic et al., 2011). Among these, presynaptic $Ca_V2.1$ channels were identified using live cell $Ca^{2+}$ imaging in rat hippocampal neurons (Glebov et al., 2017). $Ca_V2.2$ channels were also found enriched at the presynaptic active zone upon TTX treatment in rat hippocampal neurons (Glebov et al., 2017). Another study in hippocampal neurons revealed increased presynaptic neurotransmitter release after chronic activity suppression with TTX via cyclin-dependent kinase 5/calcineurin modulation of $Ca_V2.2$ channels (Kim & Ryan, 2010, 2013).

$Ca_V2$ channels rely on auxiliary subunits, particularly $\alpha_2\delta$, serving as a checkpoint for trafficking and activation of $Ca_V$ channels (Hoppa et al., 2012; Kadurin et al., 2016). Of the four different $\alpha_2\delta$ isoforms, $\alpha_2\delta$-1 is of particular interest due to its ubiquitous expression in the brain (Cole et al., 2005; Klugbauer et al., 1999). Moreover, $\alpha_2\delta$-1 plays a key role for multiple presynaptic (Brockhaus et al., 2018; Ma et al., 2018; Zhou et al., 2018) and post-synaptic functions (Eroglu et al., 2009; Risher et al., 2018), potentially independent from its association with $Ca_V$ channels (Schöpf et al., 2021).

Here, we show that $Ca_V2.2$ channels are involved in HSP in hippocampal neuronal cultures and that the over-expression of $\alpha_2\delta$-1 affects presynaptic potentiation. First, we show that $Ca_V2.2$ channels are highly expressed in both young (postnatal (P) 1 and P7) and adult (12 weeks) mouse hippocampus and cortex. Second, we demonstrate that TTX treatment increases levels of $Ca_V2.2$ channels at presynaptic boutons and their contribution to elevated $Ca^{2+}$ flux in more mature hippocampal neurons. Third, overexpression of $\alpha_2\delta$-1 results in a downregulation of $Ca_V2.2$ channel contribution to basal $Ca^{2+}$ transients. Finally, levels of endogenous $\alpha_2\delta$-1 decrease during HSP whereas the overexpression of $\alpha_2\delta$-1 prevents HSP in neurons. Together, these data suggest a key role for $Ca_V2.2$ channels and novel roles for $\alpha_2\delta$-1 during plastic adaptations of synapses.

## Methods

### Ethical approval

All experimental procedures were covered by UK Home Office licences and had local ethical approval by University College London (UCL) Bloomsbury Animal Welfare and Ethical Review Body. The study complies with the animal ethics checklist and ethical principles under which *The Journal of Physiology* operates. $Ca_V2.2\_HA^{KI/KI}$ mice were generated in a C57BL/6 background as described in Nieto-Rostro et al. (2018). Mice were housed in groups of no more than five on a 12-h:12-h light–dark cycle; food and water were available *ad libitum*. P0/P1 mice were euthanised by decapitation and older mice by cervical dislocation, except for immunohistochemistry, as described in the section below. Mice were anaesthetised with an intraperitoneal injection of pentobarbitone. Animals of both sexes were used for RT-qPCR, subcellular fractionation and hippocampal cultures.

### RT-qPCR

For gene expression studies, brains from P1, P7 and 12-week-old mice were separated into cortex, hippocampus, cerebellum and brainstem and disrupted using a rotor-stator homogeniser (Disperser T10, IKA, Staufen, Germany). Total RNA was extracted using RNeasy lipid tissue mini kit according to manufacturer's instructions. RNA concentrations were photometrically measured to reversely transcribe 5 µg RNA from each sample into complementary DNA using High-Capacity RNA-to-cDNA kit (Thermo Fisher Scientific, Waltham, MA, USA, cat. no. 4387406) (1 h at 37°C, 5 min at 95°C). For the 40-cycle qPCR (two holding stages of 50°C for 2 min and then 95°C for 10 min, followed by 40 cycles of 95°C for 15 s and 60°C for 1 min), triplicates from each sample (three different mice for each time point) were loaded into a 96-well plate with TaqMan Universal PCR Master Mix (Thermo Fisher Scientific, cat. no. 4369016) and the following TaqMan probes used (gene name: assay ID): hypoxanthine phosphoribosyltransferase 1 (*Hprt1*): Mm00446968m1; *Cacna1b*: Mm01333678m1. Optimal threshold values were defined automatically as 0.1 by Applied Biosystems 7500 Real-Time PCR software (Thermo Fisher Scientific) and used to determine the cycle threshold number ($C_T$). Results are expressed as fold change in $Ca_V2.2$ mRNA expression, given as means ± SD. Data were normalised to expression levels of internal control gene *hprt* and analysed using the $2^{-\Delta\Delta C_T}$ method (Livak & Schmittgen, 2001). To ensure sufficient amounts of RNA at time point P1 and P7, brains from two mice were pooled. For each age, three independent RNA extractions were performed and run on the same plate.

## Subcellular fractionation

Brains from P1, P7 and 12-week-old $Ca_V2.2\_HA^{KI/KI}$ mice (Nieto-Rostro et al., 2018) were dissected in buffer containing 0.32 M sucrose, 3 mM Hepes, 0.25 mM dithiothreitol, pH 7.4 and cOmplete protease inhibitor cocktail (Merck Life Sciences Ltd, Gillingham, UK, cat. no. 11836145001) and separated into cortex, hippocampus, brainstem and cerebellum. Synaptosomes were prepared as previously described (Kato et al., 2007). Crude synaptosomes were solubilised for 30 min on ice in 50 mM Tris, 150 mM NaCl, 1% Igepal, 0.5% sodium deoxycholate, 0.1% SDS and cOmplete protease inhibitor cocktail (Merck), pH 8 (Ferron et al., 2008). After centrifugation, protein concentrations were determined using the Bradford protein assay method (Bio-Rad Laboratories, Hercules, CA, USA). Samples were adjusted to the same concentration and denaturated with 100 mM dithiothreitol reducing agent and Laemmli sample buffer at 55°C for 10 min. About 20 µg of protein was loaded onto a 3−8% NuPAGE gel and proteins resolved by SDS-PAGE for 1 h 5 min at 150 V, 50 mA in running buffer. Following their separation, proteins were transferred from the gel to a polyvinylidene difluoride (PVDF) membrane using a SDi-dry transfer blot (Bio-Rad) for 10 min at 25 mV, 1 mA. The membrane was then blocked by incubation with 3% bovine serum albumin (BSA), 10 mM Tris pH 7.4, 0.5% Igepal for 1 h. Following an overnight incubation at 4°C with rat anti-haemagglutinin (HA) (1:500, monoclonal, Merck, cat. no. 11867423001) or mouse anti-glyceraldehyde phosphate dehydrogenase (GAPDH) (1:25.000, polyclonal, Thermo Fisher Scientific, cat. no. AM4300) antibodies (Abs), membranes were incubated for 1 h at room temperature with horseradish peroxidase (HRP)-coupled secondary Abs at 1:2000 for 1 h (all secondary Abs from Bio-Rad, raised in goat anti-rat HRP, cat. no. 5204-2504, and anti-mouse HRP, cat. no. 1721011). Protein bands were revealed using ECL reagent (ECL 2, Thermo Fisher Scientific) with a Typhoon 9419 phosphorimager (GE Healthcare, Chicago IL, USA) and analysed using ImageJ software (NIH, Bethesda, MD, USA). A box was drawn around each band of interest to quantify the mean grey intensity for each band. These values were then normalised to the respective GAPDH values in the same lane (containing the same sample) by division to ascertain similar protein concentrations. For ease of comparison between experiments, values were then divided by cortical protein levels on the same membrane for each age.

## Whole-cell lysate immunoblotting

Whole-cell lysates (WCL) for immunoblotting were prepared by transferring control and TTX-treated neurons at days *in vitro* (DIV) 18−22 on ice, washing them twice with phosphate-buffered saline (PBS) containing 1 mM $CaCl_2$ and $MgCl_2$ and collecting cells by scraping them in $CaCl_2/MgCl_2$ PBS containing cOmplete protease inhibitor cocktail (Merck). Lysates were cleared by centrifuging at 1000 g for 10 min at 4°C and pellets were resuspended in PBS with 1% Igepal, 0.1% SDS, 0.5% sodium deoxycholate and protease inhibitor. After brief sonication and rotation for 1 h at 4°C, cells centrifuged at $16\,000 \times g$ for 30 min at 4°C. The protein concentration was determined using Bradford protein assay (Bio-Rad) and proteins were run on a 3−8% NuPAGE gel as described above. The PVDF membrane was cut according to molecular mass markers to be able to use the mouse Abs twice, then blocked and incubated overnight at 4°C with rabbit anti-$Ca_V2.2$ II–III loop Abs at 1:500, rabbit anti-$\alpha_2\delta$-1 Abs (polyclonal, Merck, cat. no. C5105) at 1:1000 or with mouse anti-GAPDH (1:25;000) Abs. After washing with Tris-buffered saline (TBS)–0.5% Igepal, membranes were incubated for 1 h at room temperature with HRP-coupled secondary Abs at 1:2000 for 1 h. Anti-rabbit HRP secondary Abs were from Bio-Rad (cat. no. 1706515). Protein bands were revealed as described above. Values were divided by GAPDH values of the same lane and then divided by values of the control neuron band on the same membrane for ease of comparison between experiments.

## Biotinylation immunoblotting

Biotinylation experiments were adapted from our previous paper (Kadurin et al., 2016). Control and TTX-treated hippocampal neurons at DIV 18−22 were washed twice with Hanks' balanced salt solution (HBSS) containing 1 mM $CaCl_2$ and $MgCl_2$ (modified HBSS). They were then incubated with Premium Grade EZ-link Sulfo-NHS-LC-Biotin (Thermo Fisher Scientific; 1 mg/ml) in modified HBSS for 30 min at room temperature. After quenching with 200 mM glycine, cells were transferred to ice, washed twice with modified HBSS and collected by scraping. Following centrifugation at 1000 g for 10 min at 4°C, the pellet was frozen at −80°C until sufficient numbers of samples were collected for pooling (up to four independent preps). Afterwards, cells were lysed in lysis buffer containing 1% Igepal, 0.1% SDS, 0.5% sodium deoxycholate and protease inhibitor in HBSS, sonicated and rotated for 1 h at 4°C. Subsequently, protein concentrations were determined using the Bradford method as described above. Streptavidin beads were then added to a fraction of the cells while keeping some of the sample as a WCL to run on the same western blot. Streptavidin-treated samples were left on the roller at 4°C overnight and proteins revealed as described above. Protein bands were analysed in ImageJ as described above. Values were divided by GAPDH values of the WCL band for control and TTX conditions,

respectively, and normalised to control neuron values for ease of comparison between different experiments.

## Primary neuronal cultures and transfection

Primary neuronal cultures were prepared from hippocampi of P0/1 wild-type or $Ca_V2.2\_HA^{KI/KI}$ mice (Nieto-Rostro et al., 2018). After euthanasia of mice, hippocampi were dissected in ice-cold dissection medium (HBSS, Hepes 1 M, 1% w/v glucose) and dissociated in enzyme solution containing papain (HBSS, 2 mg/ml L-cysteine, 2 mg/ml BSA, 50 mg/ml glucose), papain (70 U/ml) and DNase I (1200 U/ml) in HBSS. After digestion at 37°C for 40 min, the enzyme solution was aspirated and prewarmed inactivation medium (minimum essential media (MEM), 5% v/v fetal bovine serum (FBS), 0.38% w/v glucose, 0.25% w/v BSA) was added. Hippocampi were then triturated with a P1000 micropipette with polypropylene plastic tips and cells centrifuged at room temperature at 1000 rpm for 10 min in serum medium (MEM, 5% v/v FBS, 1.38% w/v glucose). After cell pellet resuspension, cells were counted and plated at a concentration of 6800 cells/mm$^2$ on coverslips precoated with poly-D-lysine (50 µg/ml). Cells were covered with serum-free neuronal plating medium comprising Neurobasal medium, supplemented with B27 (Thermo Fisher Scientific, cat. no. 17504-044) and GlutaMAX (Thermo Fisher Scientific, cat. no. 35050-038), and kept at 37°C in 5% $CO_2$ with a medium change every 3 days.

At DIV 7, cells were transfected using Lipofectamine 2000 transfection reagent according to the manufacturer's instructions (Thermo Fisher Scientific, cat. no. 11668-030). Cells were transfected with synaptophysin-GCaMP6f (Sy-GCaMP6f) (Kadurin et al., 2016) and VAMP-mOrange 2 (VAMP-mOr2) at a ratio of 3:1 (Ferron et al., 2020). VAMP-mOrange2 was generated by replacing mCherry from pCAGGs-VAMP-mCherry by mOrange2 (gifts from Dr. Timothy Ryan), and Sy-GCaMP6f was made by replacing GCaMP3 in pCMV-SyGCaMP3 (a gift from Dr. Timothy Ryan) by GCaMP6f (Chen et al., 2013). For experiments with $\alpha_2\delta$-1 overexpression, cells were transfected with Sy-GCaMPf6, VAMP-mOr2 and either $\alpha_2\delta$-1 (pCAGGS, rat, Genbank accession number M86621) or empty vector (EV) at a ratio of 2:1:1. For immunocytochemistry experiments, cells were transfected with mCherry and $\alpha_2\delta$-1-HA and EV at 1:1:2 for $\alpha_2\delta$-1 overexpressing cells or with mCherry and EV at 1:3 for control conditions. Two hours prior to transfection, half of the cell medium was replaced with fresh medium and fresh medium was added to the previously removed medium to obtain conditioned medium. Transfection mixes for one coverslip contained 4 µg DNA in 50 µl OptiMEM and 2 µl Lipofectamine in 50 µl OptiMEM, added dropwise to the cells. After 2 h

in the incubator, medium was replaced with conditioned medium.

## Immunohistochemistry

Immunohistochemical experiments were performed using $Ca_V2.2\_HA^{KI/KI}$ and wild-type mice of 12 weeks of age. Mice were anaesthetised with an intraperitoneal injection of pentobarbitone (Euthatal, Merial Animal Health, Harlow, UK; 600 mg/kg), transcardially perfused with saline containing heparin (0.1 M) followed by perfusion with 4% w/v paraformaldehyde (PFA) in 0.1 M phosphate buffer (pH 7.4) at a flow rate of 2.5 ml/min for 5 min. Following perfusion, brains were postfixed in 4% PFA for 2 h and immersed in cryoprotective 20% sucrose overnight. Subsequently, brains were mounted in optimal cutting temperature compound and sliced into 20 µm-thick coronal sections using a cryostat and then stored at −80°C. Slices were blocked and permeabilised for 1 h at room temperature in 10% v/v goat serum (GS) and 0.2% v/v Triton X-100 in PBS and then further blocked by applying unconjugated goat F(ab) anti-mouse IgG (1:100) for 1 h at room temperature to prevent unspecific binding. Subsequently, primary Abs (rat anti-HA (as above) and anti-vGAT, rabbit polyclonal, 1:500, Synaptic Systems (Göttingen, Germany), cat. no. 131 003) were applied diluted in 5% v/v GS, 0.2% v/v Triton X-100 and 0.005% v/v $NaN_3$ overnight at 4°C. After 2 days, 4% PFA was reapplied for 30 min to ensure stabilisation of the protein–antibody complex. Sections were then incubated with the goat anti-rat and anti-rabbit secondary Abs conjugated with Alexa Fluor 488 and Alexa Fluor 594, respectively, for 2 days at 4°C (Thermo Fisher Scientific, both 1:500). After washing, sections were mounted in Vectashield Antifade Mounting Medium (Vector Laboratories, Burlingame, CA, USA). Images were taken using an LSM 780 confocal microscope (Zeiss Microscopy, Oberkochen, Germany) with a ×20 objective (89 µm optical section, pixel dwell 2.05 µs, zoom 1.8) or ×63 in super-resolution mode. After acquisition, super-resolution images underwent Airyscan processing and tile stitching using Zen software (Zeiss).

## Immunocytochemistry

For the staining of endogenous $Ca_V2.2$ channels from $Ca_V2.2\_HA^{KI/KI}$ mice, hippocampal neurons at DIV 18−22 were fixed with 1% w/v PFA–4% w/v sucrose in PBS for 5 min followed by washes in PBS. Next, neurons were blocked and permeabilised for at least 1 h at room temperature in 20% v/v horse serum, 0.1% v/v Triton X-100. Subsequently, cultures were incubated overnight at 4°C with primary Abs diluted in 10% v/v horse serum, 0.1% v/v Triton X-100 (rat

anti-HA, as above, and anti-vGluT1 (guinea pig, polyclonal, 1:1000, Synaptic Systems)). Next, cells were post-fixed with 1% w/v PFA–4% w/v sucrose in PBS for 5 min at room temperature and incubated with respective donkey Alexa Fluor secondary Abs applied for 1 h diluted at 1:500. Following washes in PBS, cells were mounted in Vectashield and examined using confocal or super-resolution Airyscan mode imaging on a Zeiss LSM 780 confocal microscope with ×63 objective (1768 × 1768 pixels) as z-stacks (0.173 µm optical section). To quantify the signal intensity of $Ca_V2.2\_HA$, up to 75 regions of interest of 2 µm in diameter were manually chosen based on vGluT1 and $Ca_V2.2\_HA$ colocalisation on a single plane and the intensity measured in ZEN (Zeiss, version 5). Values from TTX boutons were normalised to control values. Analysis was performed blind and randomised.

For the $\alpha_2\delta$-1-HA staining, hippocampal cells were fixed with 4% w/v PFA–4% w/v sucrose in PBS for 5 min followed by washes in PBS. Next, cells were blocked and permeabilised for at least 1 h at room temperature in 20% v/v goat serum–0.3% v/v Triton X-100. Subsequently, cultures were incubated overnight at 4°C with rat anti-HA Abs at 1:200 (Merck) and guinea-pig anti-red fluorescent protein (RFP) Abs at 1:500 (Synaptic Systems) diluted in 10% v/v goat serum–0.3% v/v Triton X-100 (Ab solution). Next, cells were washed with PBS, and goat anti-rat Alexa Fluor 488 and goat anti-guinea-pig Alexa Fluor 594 secondary Abs were applied for 1 h diluted at 1:500 in Ab solution. After washing with PBS, cells were mounted using Vectashield. Images were acquired using the confocal mode of a Zeiss LSM 780 confocal microscopes with a ×40 oil-immersion objective in 8-bit mode. Images were taken as tile (2 × 2 each consisting of 1024 × 1024 pixels) and z-stack scans (0.28 µm optical section) with a pixel dwell of 2.05 µs.

### Live cell $Ca^{2+}$ imaging

$Ca^{2+}$ imaging experiments were performed as described in a previous paper (Ferron et al., 2020). Plated cells on 22 $mm^2$ coverslips were transferred to a laminar-flow perfusion and stimulation chamber (imaging chamber with field stimulation series 20, Warner Instruments, Holliston, MA, USA) and mounted on an epifluorescence microscope (Zeiss Axiovert 200M) under continuous perfusion at 23°C at 0.5 ml/min with $Ca^{2+}$ perfusion buffer containing (in mM) 119 NaCl, 2.5 KCl, 2 $CaCl_2$, 2 $MgCl_2$, 25 Hepes (buffered to pH 7.4) and 30 glucose. In order to suppress postsynaptic activity, 10 µM 6-cyano-7-nitroquinoxaline-2,3-dione (CNQX, Sigma) and 50 µM D,L-2-amino-5-phosphonovaleric acid (AP5, Sigma) were included. For some experiments, the irreversible $Ca_V2.2$ channel inhibitor $\omega$-conotoxin GVIA (ConTx; 1 µM, Alomone Laboratories, Jerusalem,

Israel) was applied for 2 min before stimulation under continuous perfusion. To ascertain whether any observed reduction in fluorescence was due to the incubation period or due to bleaching during re-stimulation, control experiments were performed with normal imaging medium applied for 2 min instead of ConTx and cells were re-stimulated, which resulted in a reduction of 8.8 ± 9.0% during one AP stimulation. All values shown have been adjusted for this reduction.

Homeostatic presynaptic plasticity was induced by incubating cells with 1.5 µM tetrodotoxin (TTX) for 48 h prior to imaging (Zhao et al., 2011). Before imaging, cells were incubated in $Ca^{2+}$ perfusion buffer for 20 min to wash out TTX. Images were acquired with an Andor iXon+ (model DU-897U-CS0-BV) back-illuminated EMCCD camera using OptoMorph software (Cairn Research, Faversham, UK) with LEDs as light sources (Cairn Research). Fluorescence excitation and collection was done through a ×40 1.3 NA Fluar Zeiss objective using 450/50 nm excitation and 510/50 nm emission and 480 nm dichroic filters (for Sy-GCaMP6f) and a 572/35 nm excitation and low-pass 590 nm emission and 580 nm dichroic filters (for VAMP-mOr2). APs were evoked by passing 1 ms current pulses via platinum electrodes. Transfected boutons were selected for imaging by stimulating neurons with trains of six APs at 33 Hz using a Digitimer D4030 and DS2 isolated voltage stimulator (Digitimer Ltd, Welwyn Garden City, UK). To measure calcium responses, neurons were stimulated with a single AP (repeated at least five times with 30 s time intervals to improve signal-to noise ratio) and then with 10 APs at 10 Hz. Synaptic boutons were identified by VAMP-mOr2 expression and defined as functional based on responsiveness to stimulation with 200 APs at 10 Hz. From each coverslip, a maximum of three fields of view were recorded. When ConTx was applied, only one field of view was imaged per coverslip. To get a baseline value of fluorescence, 20 frames were recorded before stimulation ($F_0$). Images were acquired at 100 Hz and 7 ms exposure time and up to 75 putative synaptic boutons within the image field were selected for analysis using a 2 µm region of interest (ROI) and analysed in ImageJ using a custom-written plugin (http://www.rsb.info.nih.gov/ij/plugins/time-series.html). Data were background adjusted and changes calculated as change of fluorescence intensity over baseline fluorescence before stimulation ($\Delta F/F_0$). For presentational purposes, images were adjusted for brightness and contrast.

### Statistics

Data are given as means ± SD with the number of independent experiments (*n*) and statistical test used specified for each figure. Results were considered

significant with a *P*-value <0.05. Data were analysed and graphs generated with GraphPad Prism 9 (GraphPad Software Inc., San Diego, CA, USA): ns: not significant, $P > 0.05$; *$P \leq 0.05$; **$P \leq 0.01$; ***$P \leq 0.001$.

## Results

### Ca$_V$2.2 channels in the immature and mature brain

In the mammalian brain, both Ca$_V$2.1 and Ca$_V$2.2 channels provide the main sources of Ca$^{2+}$ influx at pre-synaptic terminals. The abundance of Ca$_V$2 channels at the presynapse depends on specific synapse needs and therefore varies between different ages, brain regions, synaptic type and activity (Dolphin & Lee, 2020).

To determine the relative expression levels of Ca$_V$2.2 channels in different brain regions at three different ages, RT-qPCR experiments were performed (Fig. 1*A*–*C*). Analysis of Ca$_V$2.2 mRNA expression in cortex, hippocampus, cerebellum and brainstem revealed similar expression levels of Ca$_V$2.2 mRNA in young brains at P1, whereas at P7 and in the adult brain, the expression of Ca$_V$2.2 channel mRNA was significantly higher in the cortex compared to cerebellum and brainstem (P7: hippocampus, $0.91 \pm 0.19$; cerebellum, $0.68 \pm 0.1$; brainstem, $0.62 \pm 0.11$; adult: hippocampus, $1.15 \pm 0.09$; cerebellum, $0.67 \pm 0.09$; brainstem, $0.23 \pm 0.05$; Fig. 1*A*–*C*, one-way ANOVA, Bonferroni *post hoc* test). At all ages, levels were similar in cortex and hippocampus. In parallel, to correlate Ca$_V$2.2 mRNA levels to endogenous Ca$_V$2.2 protein expression, brains from transgenic knock-in mice with an exofacial haemagglutinin (HA) tag on Ca$_V$2.2 channels (Ca$_V$2.2_HA$^{KI/KI}$; Nieto-Rostro et al., 2018) were used for quantitative immunoblotting of synaptosomes (Fig. 1*D* and *E*). Immunoblots from brain synaptosomes show that the distribution of Ca$_V$2.2 channels follows a trend similar to mRNA levels. At P1 and P7 the distribution of Ca$_V$2.2 channels did not vary much between the different regions, whereas in the adult brain, levels of Ca$_V$2.2 channels were higher in the cortex compared to cerebellum and brainstem (relative to 1 in cortex; levels were $0.88 \pm 0.20$ in hippocampus, $0.09 \pm 0.09$ in cerebellum and $0.14 \pm 0.11$ in brainstem; Fig. 1*D*–*I*, one-sample two-tailed *t* test). Like in the mRNA experiments, protein levels of cortex and hippocampus were similar at all ages. Figure 1*J*–*L* shows immunostaining of Ca$_V$2.2_HA channels around the somata of hippocampal CA1 cells from Ca$_V$2.2_HA$^{KI/KI}$ mice.

### Induction of homeostatic synaptic plasticity in mature hippocampal neurons

Silencing neuronal activity has been shown to induce compensatory changes at both the pre- and postsynapse (Turrigiano, 2008). To specifically examine changes in presynaptic Ca$^{2+}$ transients, crucial for neurotransmitter release and therefore presynaptic strength, hippocampal neurons were transfected with genetically encoded Ca$^{2+}$ indicator GCaMP6f coupled to presynaptic synaptophysin (Sy-GCaMP6f; Kadurin et al., 2016; Fig. 2). Neurons were stimulated with 1 and 10 APs (Fig. 2*A* and *B*). Co-expression of pH indicator mOrange 2 (mOr2) coupled to synaptic vesicle associated membrane protein (VAMP) allowed for targeted analysis of Ca$^{2+}$ transients at neurotransmitter-releasing synaptic boutons (Fig. 2*C*) (Ferron et al., 2020). For analysis of Ca$^{2+}$ transients, only the responses from releasing (orange line, Fig. 2*F*) boutons were included. Figure 2*D* and *E* shows changes in fluorescence in $\Delta F/F_0$ of releasing (open circles) and of non-releasing (filled circles) boutons after stimulation with 1 (Fig. 2*D*) and 10 (Fig. 2*E*) APs.

To induce homeostatic changes at the synapse, neurons were incubated with TTX for 48 h and Ca$^{2+}$ transients were measured at two ages: between day *in vitro* (DIV) 14 and 15 and between DIV 18 and 22 (Fig. 3). In younger cultures, at DIV 14−15, no significant changes in Ca$^{2+}$ transient amplitudes were detected between control and TTX-treated neurons ($0.032 \pm 0.008$ in control cells and $0.042 \pm 0.02$ after TTX treatment, paired *t* test, $P = 0.15$; Fig. 3*A*). However, in more mature cultures at DIV 18−22, measured Ca$^{2+}$ transient amplitudes were much larger following TTX treatment ($0.024 \pm 0.008$ in control neurons and $0.038 \pm 0.008$ in TTX-treated neurons, paired *t* test, $P = 0.007$; Fig. 3*B*). Similarly, during stimulation with 10 APs, control and TTX-treated boutons showed similar Ca$^{2+}$ peak amplitudes at DIV 14−15 ($0.52 \pm 0.12$ in control neurons and $0.47 \pm 0.05$ in TTX-treated neurons, paired *t* test, $P = 0.38$; Fig. 3*C*). However, in older hippocampal cultures, chronic TTX application resulted in increased presynaptic peak Ca$^{2+}$ amplitudes compared to control neurons ($0.35 \pm 0.1$ in control boutons and $0.52 \pm 0.13$ in TTX-treated boutons, paired *t* test, $P = 0.01$; Fig. 3*D*).

### Increased levels of Ca$_V$2.2 channels contribute to larger Ca$^{2+}$ transients during HSP

At presynaptic terminals, a combination of different Ca$_V$2 channels mediate Ca$^{2+}$ influx. To examine the contribution of Ca$_V$2.2 channels during HSP, cells were stimulated before and after the application of the Ca$_V$2.2 channel-specific inhibitor $\omega$-conotoxin GVIA (ConTx; Fig. 4*A*–*D*). Figure 4*A* shows Sy-GCaMP6f Ca$^{2+}$ transients from control and TTX-treated boutons before (grey and magenta, respectively) and after the application of ConTx (Control + ConTx in red and TTX + ConTx in orange). At DIV 14−15, Ca$_V$2.2 channel contribution

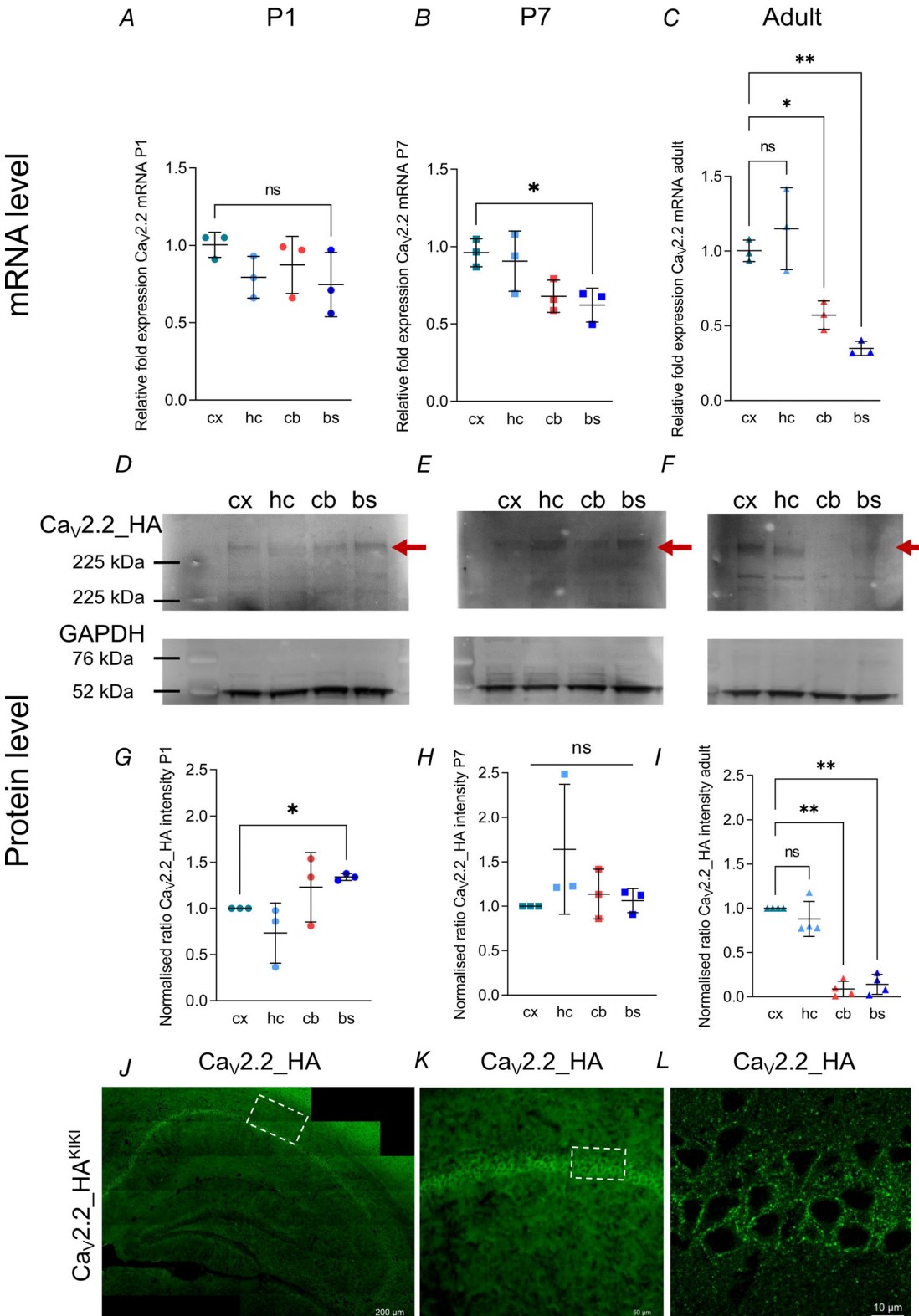

**Figure 1. Differential expression of Ca$_V$2.2 mRNA and protein levels in different brain regions with most Ca$_V$2.2 in the adult cortex and hippocampus**

*A*, at P1, mRNA levels of Ca$_V$2.2 were similar in all four brain regions (cortex (cx) hippocampus (hc), cerebellum (cb) and brainstem (bs)). One-way ANOVA, $F(3,6) = 1.975$, $P = 0.22$, Bonferroni *post hoc* test; cx *vs.* hc, $P = 0.43$; cx *vs.* cb, $P > 0.1$; cx *vs.* bs, $P = 0.78$; $n = 3$ independent experiments with brains from two pups for each

experiment. *B*, at P7, qPCR experiments reveal higher expression levels of Ca$_V$2.2 mRNA in the cortex compared to the cerebellum and brainstem. One-way ANOVA, $F(3,6) = 13.9$, $P = 0.004$, Bonferroni *post hoc* test, cx *vs.* hc, $P > 0.1$; cx *vs.* cb, $P = 0.01$; cx *vs.* bs, $P = 0.005$; $n = 3$ independent experiments with brains from two pups for each experiment. *C*, in adult brains (12 weeks old), Ca$_V$2.2 mRNA levels were significantly reduced in cerebellum and brainstem compared to cortex and hippocampus. $n = 3$ independent experiments, one-way ANOVA, $F(3,8) = 18.3$, $P = 0.0006$, Bonferroni *post hoc* test, cx *vs.* hc, $P = 0.8$; cx *vs.* cb, $P = 0.02$; cx *vs.* bs, $P = 0.002$; ns, not significant. Each $n$ in *A–C* was assayed in triplicate. All fold changes are relative to Ca$_V$2.2 mRNA levels in the cortex and respective to hypoxanthine-guanine phosphoribosyltransferase (HPRT) mRNA. Data are shown as means $\pm$ SD. *D–F*, immunoblots of synaptosomes from Ca$_V$2.2_HA$^{KI/KI}$ mice showing the expression of Ca$_V$2.2_HA (red arrow; top) and glyceraldehyde 3-phosphate dehydrogenase (GAPDH, bottom) for P1 (left), P7 (middle) and adult (right) mice. The molecular mass of Ca$_V$2.2_HA is $261.0 \pm 1.2$ kDa, determined from molecular mass markers. *G*, protein band quantification reveals similar levels of Ca$_V$2.2_HA in cortex, hippocampus, cerebellum and brainstem from P1 Ca$_V$2.2_HA$^{KI/KI}$ mice, with higher levels in brainstem compared to cortex. For $n = 3$ independent experiments with eight mice pooled for each set, data were normalised to the cx in each experiment, and differences compared to cx determined by one-sample two-tailed $t$ test compared to a theoretical mean of 1; cx *vs.* hc, $P = 0.29$; cx *vs.* cb, $P = 0.40$; cx *vs.* bs, $P = 0.004$; *H*, similar levels of Ca$_V$2.2_HA in cortex, hippocampus, cerebellum and brainstem from P7 Ca$_V$2.2_HA$^{KI/KI}$ mice. For $n = 3$ independent experiments with four mice pooled for each set, data were normalised to the cx in each experiment, and differences compared to cx determined by one-sample two-tailed $t$ test compared to a theoretical mean of 1; cx *vs.* hc, $P = 0.29$; cx *vs.* cb, $P = 049$; cx *vs.* bs, $P = 0.51$. *I*, in adult mice, Ca$_V$2.2_HA levels are higher in the cortex compared to cerebellum and brainstem. For $n = 4$ independent experiments with one mouse per experiment, data were normalised to the cx in each experiment and differences compared to cx determined by one-sample two-tailed $t$ test compared to a theoretical mean of 1. cx *vs.* hc, $P = 0.31$; cx *vs.* cb, $P = 0.0002$; cx *vs.* bs, $P = 0.0006$. For *G–I*, all values are normalised to cortex Ca$_V$2.2_HA and to respective GAPDH levels of each gel, and data are shown as means $\pm$ SD. *J*, Ca$_V$2.2_HA immunolabelling in the hippocampus of adult Ca$_V$2.2_HA$^{KI/KI}$ mice visualised using anti-HA antibodies (green). $\times$20 super-resolution tile scan; scale bar, 200 μm. *K*, Ca$_V$2.2_HA (green) signal in CA1 area of the hippocampus, $\times$20 confocal imaging; scale bar, 50 μm. *L*, Ca$_V$2.2_HA (green) $\times$63 image of CA1 somata of Ca$_V$2.2_HA$^{KI/KI}$ mice in super-resolution, maximum intensity projection of z-stack with 0.197 μm optical sections; scale bar, 10 μm. [Colour figure can be viewed at wileyonlinelibrary.com]

to Ca$^{2+}$ transients during one AP was not statistically different between control and TTX-treated neurons. In control and TTX-treated neurons, $29.53 \pm 24.08\%$ and $49.23 \pm 21.07\%$ of total Ca$^{2+}$ influx was mediated by Ca$_V$2.2 channels, respectively (unpaired $t$ test, $P = 0.08$; Fig. 4*B*). Conversely, during HSP, in more mature neurons, the contribution of Ca$_V$2.2 channels to overall Ca$^{2+}$ influx increased. Traces of Sy-GCaMP6f fluorescence before ConTx application (grey for control and blue for TTX-treated boutons) and after toxin application (red and orange) are shown in Fig. 4*C*. At DIV 18−22 the contribution of Ca$_V$2.2 to Ca$^{2+}$ flux rose from $28.6 \pm 25.12\%$ in control neurons to $55.3 \pm 14.79\%$ in potentiated hippocampal neurons treated with TTX (unpaired $t$ test, $P = 0.007$; Fig. 4*D*). Moreover, immuno-blotting of whole cell lysates (WCL) of hippocampal neurons at DIV 18−22 showed a $37.6 \pm 0.03\%$ increase in levels of Ca$_V$2.2 channels after the induction of HSP with TTX (one-sample two-tailed $t$ test, $P = 0.002$; Fig. 4*E* and *F*). To determine whether this increase was associated with presynaptic boutons, we used cultured hippocampal neurons from transgenic Ca$_V$2.2_HA$^{KI/KI}$ mice to visualise endogenous Ca$_V$2.2 channels *in vitro* (Fig. 4*G*). Quantification of Ca$_V$2.2_HA intensity associated with puncta positive for the presynaptic marker vesicular glutamate transporter 1 (vGluT1) revealed an increase of almost 40% of Ca$_V$2.2_HA channel levels in neurons treated with TTX (control normalised mean intensity $100 \pm 39.19\%$ and TTX mean intensity $139.9 \pm 56.53\%$; unpaired $t$ test $P = 0.006$; Fig. 4*G–I*).

### $\alpha_2\delta$-1 overexpression does not change Ca$^{2+}$ transients resulting from one AP stimulation and prevents HSP

The Ca$_V$ $\alpha_2\delta$ subunit is emerging as an important regulator of presynaptic organisation and function (Ferron et al., 2018; Hoppa et al., 2012; Schöpf et al., 2021). Hence, we investigated the effect of $\alpha_2\delta$-1 overexpression on Ca$^{2+}$ transient amplitudes and the Ca$_V$2.2 channel contribution to Ca$^{2+}$ flux in hippocampal neurons aged DIV 18−22 (Fig. 5).

The overexpression of $\alpha_2\delta$-1 is shown in Fig. 5*A* using anti-HA Abs against tagged $\alpha_2\delta$-1_HA; no background staining is visible in the EV control neurons. Comparison of Ca$^{2+}$ transients after one AP stimulation showed no changes in Ca$^{2+}$ influx (paired $t$ test; $P = 0.18$; Fig. 5*B* and *C*) with similar fluorescence profiles in the EV control (grey, $0.025 \pm 0.006$) and the $\alpha_2\delta$-1-overexpressing boutons (green, $0.028 \pm 0.008$). Since we found no changes in Ca$^{2+}$ transient amplitudes in $\alpha_2\delta$-1-overexpressing neurons, we then applied ConTx to assess if the contribution of Ca$_V$2.2 channels to Ca$^{2+}$ transients was altered. ConTx application revealed that in $\alpha_2\delta$-1-overexpressing neurons, the contribution of Ca$_V$2.2 channels to Ca$^{2+}$ transients after one AP stimulation decreased to $28.1 \pm 16.38\%$ compared to $47.1 \pm 20.65\%$

in EV-transfected neurons (unpaired $t$ test; $P = 0.05$; Fig. 5$D$).

Not only is the $\alpha_2\delta$-1 subunit important for the trafficking and function of Ca$_V$2.2 channels, but it is also emerging as an important trans-synaptic regulator with several described functions independent from its association with Ca$_V$2 channels (Schöpf et al., 2021). To elucidate if the induction of HSP changes the surface distribution of endogenous $\alpha_2\delta$-1, biotinylation assays were performed using control and TTX-treated neurons at DIV 18−22 (Fig. 6$A$ and $B$). The hippocampal neurons potentiated with TTX revealed a decrease of surface (biotinylated fraction) and total endogenous $\alpha_2\delta$-1 (WCL fraction; decrease to $88.61 \pm 0.02\%$ and to $86.8 \pm 0.08\%$, respectively; one-sample two-tailed $t$ test, WCL $P = 0.007$ and surface fraction $P = 0.04$; Fig. 6$A$ and $B$). To then

further assess the role of $\alpha_2\delta$-1 for HSP, we overexpressed $\alpha_2\delta$-1 in neurons and recorded Ca$^{2+}$ transients after incubation with TTX (Fig. 6$C$). In neurons overexpressing $\alpha_2\delta$-1, no increases in Sy-GCaMP6f Ca$^{2+}$ transient amplitudes were induced by treatment with TTX, whereas Ca$^{2+}$ transients in EV untreated control neurons were larger after TTX treatment, similar to values shown in Fig. 3 ($0.023 \pm 0.005$ in EV untreated neurons and $0.04 \pm 0.02$ in TTX-treated EV neurons; $0.027 \pm 0.006$ in $\alpha_2\delta$-1-overexpressing neurons and $0.029 \pm 0.009$ in TTX-treated $\alpha_2\delta$-1-overexpressing neurons, one-way ANOVA, $P$ EV $= 0.0008$ and $P$ $\alpha_2\delta$-1 $= 0.99$; Fig. 6$C$). These findings could have important implications for the role of $\alpha_2\delta$-1 in regulating both Ca$_V$2.2 channel plasticity and other mechanisms involved in HSP.

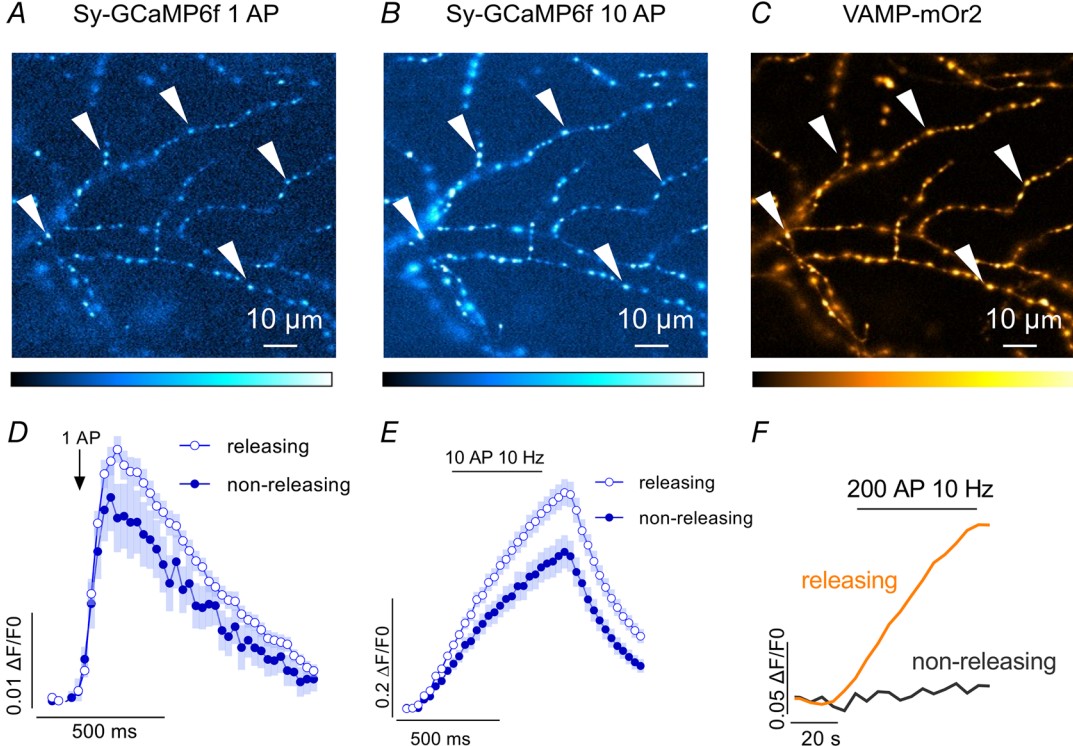

**Figure 2. Monitoring presynaptic Ca$^{2+}$ transients in hippocampal neurons with Sy-GCaMP6f and VAMP-mOr2**
*A*, images showing the expression of Sy-GCaMP6f in putative boutons during stimulation with one AP (arrowheads pointing at exemplary boutons). *B*, changes in Sy-GCaMP6f fluorescence after stimulation with 10 APs were used to identify responding boutons (arrowheads). *C*, VAMP-mOr2 fluorescence after stimulation with 200 APs at 10 Hz allowed identification of functional vesicle-releasing synapses based on an increase in fluorescence following the increase in pH to which it is exposed during vesicle fusion. *D*, up to 75 ROIs were selected per field of view to show changes in fluorescence over baseline ($\Delta F/F_0$) of averages of 5−8 repeats of one AP stimulation. Responses from releasing (blue open circles) and non-releasing (blue filled circles) boutons were distinguished based on VAMP-mOr2 responses. *n* for releasing boutons = 57 fields of view, *n* for non-releasing boutons = 34 fields of view. *E*, responses from releasing (blue open circles) and non-releasing (blue filled circles) boutons after stimulation with 10 Aps. *n* for releasing boutons = 52 fields of view, *n* for non-releasing boutons = 32 fields of view. *F*, increase in VAMP-mOr2 fluorescence (orange line) after stimulation with 200 APs was used to identify releasing boutons (orange line) and non-releasing boutons (black line). *n* for releasing boutons = 23, *n* for non-releasing boutons = 11. Based on this, responses to 1 and 10 APs were categorised into releasing and non-releasing boutons. [Colour figure can be viewed at wileyonlinelibrary.com]

## Discussion

Presynaptic Ca$_V$2 channels play a pivotal role in synaptic transmission by mediating fast neurotransmitter exocytosis via influx of Ca$^{2+}$ into the active zone. Here, we combine gene expression studies, immunoblotting, immunocytochemistry and live cell Ca$^{2+}$ imaging to show (i) the role of Ca$_V$2.2 channels for presynaptic Ca$^{2+}$ flux in hippocampal cultures, (ii) upregulation of Ca$_V$2.2 channels mediates increased Ca$^{2+}$ flux during HSP, (iii) HSP downregulates endogenous $\alpha_2\delta$-1 subunits at synapses in hippocampal cultures, and (iv) over-expression of $\alpha_2\delta$-1 decreases the contribution of Ca$_V$2.2

to presynaptic Ca$^{2+}$ flux and abolishes the effect of TTX to elevate Ca$^{2+}$ transients.

### Ca$_V$2.2 channels in the adult hippocampus

We show high levels of Ca$_V$2.2 channel protein and mRNA expression in mouse cortex and hippocampus throughout development, persisting into adulthood. While younger mice showed relatively similar expression of Ca$_V$2.2 mRNA levels across brain regions, adult cortex and hippocampus had higher levels compared to cerebellum and brainstem. This finding was supported

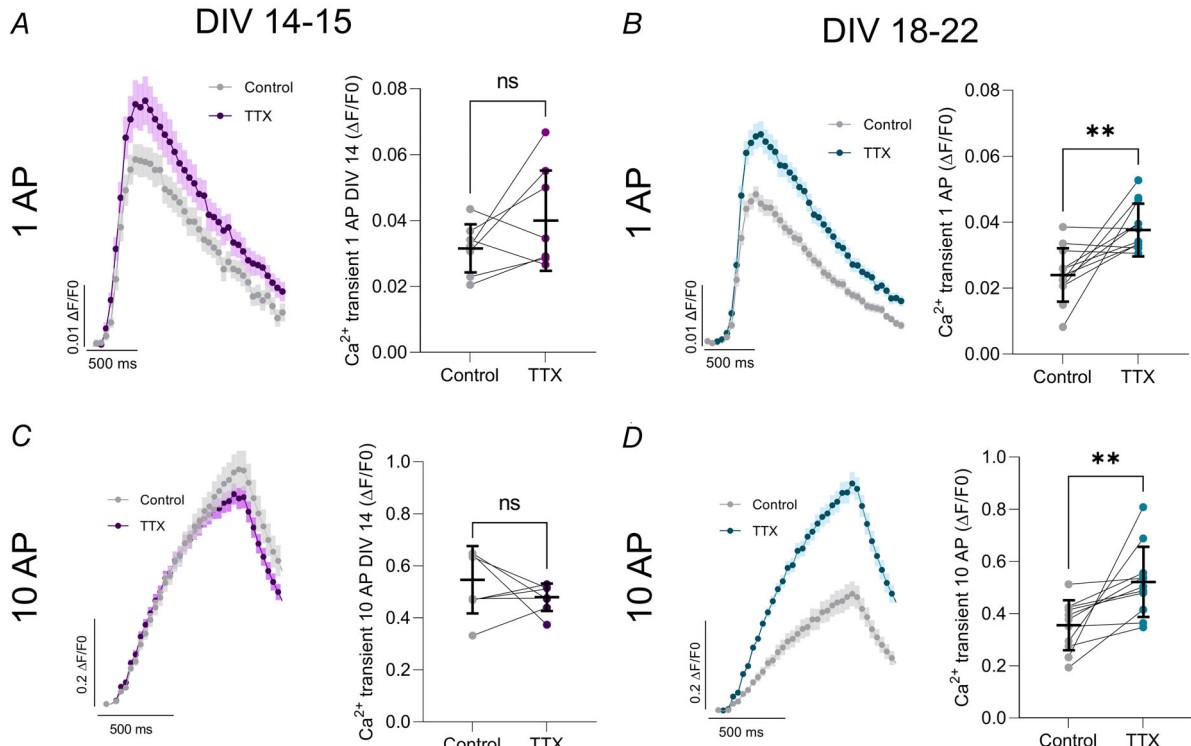

**Figure 3. TTX treatment increases presynaptic Ca$^{2+}$ transient amplitudes in more mature mouse hippocampal boutons**

*A*, Sy-GCaMP6f fluorescence changes in functionally releasing presynaptic boutons after stimulation with one AP in control (grey) and TTX (magenta) conditions at DIV 14−15 (left panel) shown as averaged traces with SEM. At DIV 14−15, no significant differences were observed between presynaptic Ca$^{2+}$ transient amplitudes in control (grey) and TTX-treated (magenta) boutons (right panel). *n* = 7 biological replicates, two-tailed paired *t* test *P* = 0.15; *n* corresponds to independent experiments and data are shown as means ± SD (black). *B*, Sy-GCaMP6f fluorescence changes at DIV 18−22 in TTX (blue)-treated presynaptic terminals and control (grey) untreated terminals during one AP stimulation. Traces are averaged values and are shown as means ± SEM (left panel). At DIV 18−22, TTX treatment (blue) induced an increase in presynaptic Ca$^{2+}$ transient amplitudes compared to control boutons (grey; right panel). *n* = 11 biological replicates, two-tailed paired *t* test, *P* = 0.007; *n* corresponds to independent experiments and data are shown as means ± SD (black). *C*, Sy-GCaMP6f fluorescence at DIV 14−15 in control (grey) and TTX-treated (magenta) presynaptic boutons during stimulation with 10 APs. Traces are averaged values and are shown as means ± SEM (left panel). Comparison of the Ca$^{2+}$ transient amplitudes reveal similar levels for control and TTX-treated neurons (right panel). *n* = 7 biological replicates, two-tailed paired *t* test, *P* = 0.38; *n* corresponds to independent experiments and data are shown as mean ± SD (black). Averaged Sy-GCaMP6f traces of control (grey) and TTX-treated (blue) boutons (left panel) shown ± SEM (left panel). *D*, in more mature cultures at DIV 18−22, Sy-GCaMP6f fluorescence increased after TTX treatment (blue) compared to control boutons (grey). *n* = 11 biological replicates, two-tailed paired *t* test, *P* = 0.01; *n* corresponds to independent experiments and data are shown as means ± SD (black). [Colour figure can be viewed at wileyonlinelibrary.com]

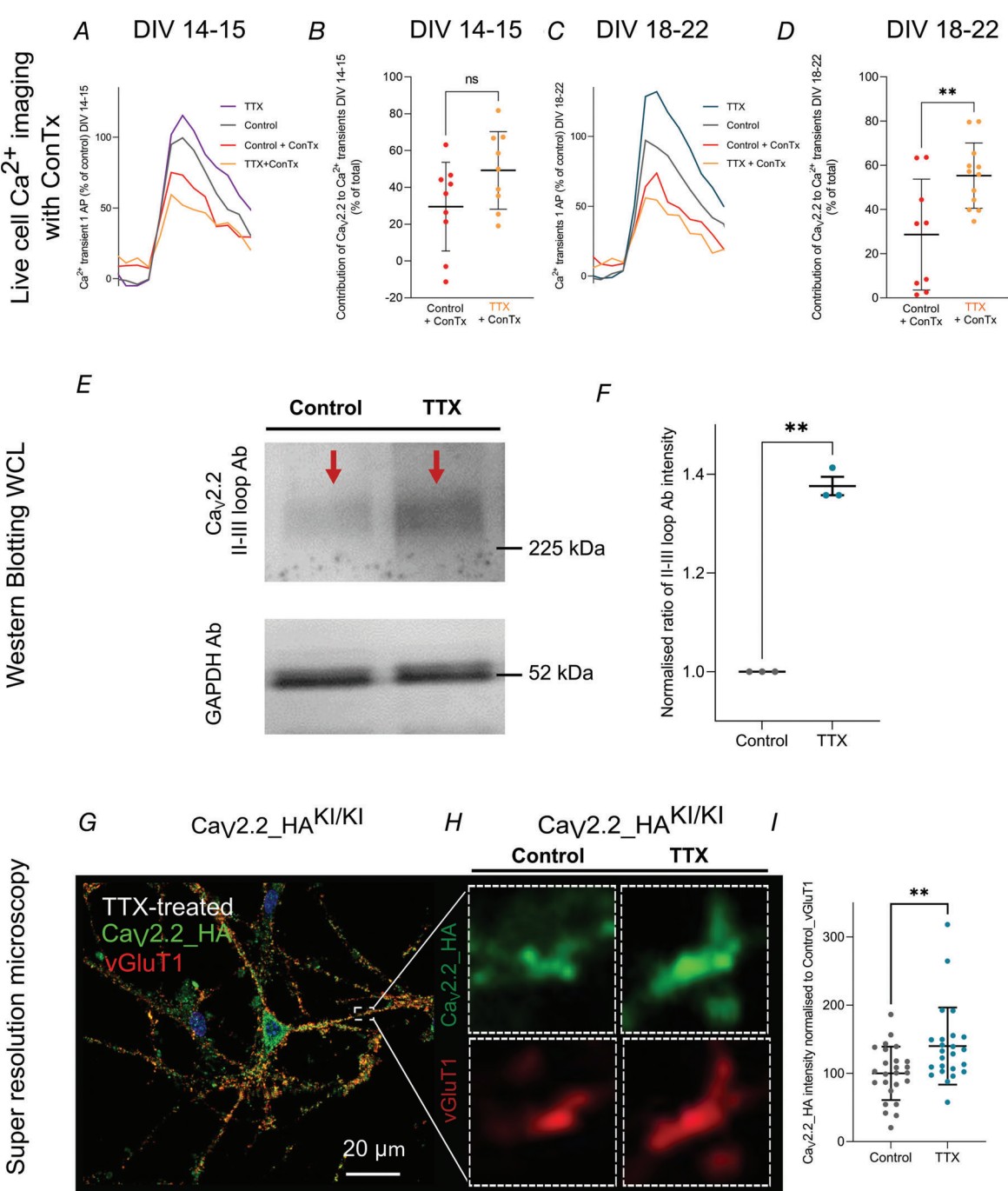

**Figure 4. Ca$_V$2.2 channels mediate increased Ca$^{2+}$ transients during HSP at DIV 18−22**

*A*, averaged Ca$^{2+}$ transients from control (grey) and TTX-treated (magenta) boutons at DIV 14−15 and post-ConTx application (red traces for control + ConTx and orange traces for TTX + ConTx). Values are normalised to control neurons. *n* for control = 9, *n* for TTX = 9. *B*, at DIV 14−15, contribution of Ca$_V$2.2 channels to Ca$^{2+}$ transient amplitudes after electrical stimulation with one AP was not significantly different from untreated neurons (red data points) and after TTX incubation (orange data points). Two-tailed unpaired *t* test, *P* = 0.083, *n* for control = 9, *n* for TTX = 9; *n* corresponds to fields of view from three independent experiments and data are shown as means ± SD; values are normalised to their respective control pre-toxin incubation. *C*, averaged Sy-GCaMP6f traces for control and TTX-treated presynaptic boutons in grey and blue, respectively, and decreased traces after application of ConTx in control neurons (red) and TTX-treated neurons (orange). Values are normalised to control neurons; *n* for control = 9, *n* for TTX = 12. *D*, at DIV 18−22, Ca$_V$2.2 channels contribute more to Ca$^{2+}$ flux after the induction of HSP with TTX. Two-tailed unpaired *t* test, *P* = 0.007; *n* for control = 9, *n* for TTX = 12; *n* corresponds to fields of view from three independent experiments and data are shown as means ± SD; values are normalised to their respective

control pre-toxin incubation. *E*, representative immunoblots using Abs against the II–III loop of Ca$_V$2.2 channels in WCL of control neurons and TTX-treated neurons (red arrowhead, top gel) at DIV 18–22. Values were normalised to the intensity of GAPDH for control and TTX neurons, respectively (bottom gel). *F*, quantification of the intensity of Ca$_V$2.2 II–III loop band of three independent experiments reveals a stronger intensity after the induction of HSP with TTX (blue data points) normalised to control, untreated neurons (grey data points). One sample two-tailed *t* test compared to theoretical mean of 1, *P* = 0.002; *n* = 3 independent experiments, averaged duplicates for each experiment. *G*, Airyscan image of TTX-treated hippocampal neuron from Ca$_V$2.2_HA$^{KI/KI}$ mouse at DIV 21 stained with anti-HA Abs (green) and anti-vGluT1 Abs (red) to identify presynaptic terminals. Magnification ×63; scale bar, 20 µm. *H*, 3 × 3 µm subset images from white box in *G*. Control boutons (left panels) and TTX-treated (right panels) with Ca$_V$2.2_HA in green and vGluT1 in red. *I*, graph showing increased Ca$_V$2.2_HA intensity in TTX-treated neurons compared to control neurons as means ± SD. *n* for control neurons = 25, *n* for TTX-treated neurons = 24, from three independent experiments; for each neuron, up to 75 ROIs were selected; data were normalised to the averaged control value; two-tailed unpaired *t* test, *P* = 0.006. [Colour figure can be viewed at wileyonlinelibrary.com]

by protein immunoblot studies using synaptosomal fractionations of brains from Ca$_V$2.2_HA$^{KI/KI}$ mice. We also, for the first time, visualise Ca$_V$2.2_HA channels in the hippocampus using the Ca$_V$2.2_HA$^{KI/KI}$ mouse model.

One of the key questions regarding Ca$_V$2.2 channels is how they are distributed and regulated in different brain regions and at different synapses. Initial work on this examined the calyx of Held, a large auditory relay synapse in the brainstem. Before hearing onset in young mice, synaptic transmission is mediated by loosely coupled Ca$_V$2.1 and Ca$_V$2.2 channels, whereas in the mature synapse Ca$^{2+}$ flux is mediated by tightly coupled Ca$_V$2.1 channels (Fedchyshyn & Wang, 2005; Iwasaki & Takahashi, 1998; Iwasaki et al., 2000). This nanodomain coupling of Ca$_V$2.1 channels allows rapid and temporally precise glutamate release, required for auditory processing. Although the developmental shift from Ca$_V$2.2 to Ca$_V$2.1 channels has also been described in the neocortex (Bornschein et al., 2019), hippocampus (Scholz & Miller, 1995) and cerebellum (Miki et al., 2013), our findings do not provide evidence for a downregulation of Ca$_V$2.2 channels in the adult hippocampus and cortex. Nevertheless, there may be a parallel upregulation of Ca$_V$2.1 channels that needs to be assessed further. There is evidence that Ca$_V$2.2 channels at least partially mediate presynaptic Ca$^{2+}$ flux in the adult cortex and hippocampus (Brockhaus et al., 2019; Cao & Tsien, 2010; Ermolyuk et al., 2013; Ferron et al., 2020; Hoppa et al., 2012; Wheeler et al., 1996). Highly plastic synapses, such as in the hippocampus, may use a combination of Ca$_V$2.1, Ca$_V$2.2 and Ca$_V$2.3 channels, each providing the synapse with distinct coupling properties, activity-dependent facilitation and modulation of Ca$_V$2 channels. This may enable dynamic changes in synaptic output depending on synaptic activity (Dolphin & Lee, 2020; Eggermann et al., 2011). The hippocampal mossy fibre pathway, for example, was shown to rely on microdomain coupling of Ca$_V$2.2 channels for presynaptic plasticity (Vyleta & Jonas, 2014). Notably, the shape of APs in different brain regions and neuron types (e.g. narrow APs in interneurons and in the calyx of Held *versus* broader APs at the

hippocampal presynapse; Bean, 2007) might be another factor determining which Ca$_V$2 channel subtype is predominantly activated. The mechanisms underlying the differential distribution of Ca$_V$2.1 and Ca$_V$2.2 channels remain unclear.

### Increased levels of Ca$_V$2.2 channels contribute to increased Ca$^{2+}$ flux during HSP

Our observation that about 30% of Ca$^{2+}$ influx occurs via Ca$_V$2.2 channels in mature mouse hippocampal neurons is similar to previous findings (Brockhaus et al., 2019; Cao & Tsien, 2010), which is suggestive of an important role for Ca$_V$2.2 channels in synaptic transmission in mature neurons. Interestingly, ConTx application revealed similar levels of Ca$_V$2.2 channel contribution to calcium flux at both DIV 14−15 and DIV 18−22, further evidence against a developmental downregulation of Ca$_V$2.2 channels as neurons mature. Chronic silencing of neuronal activity with TTX in hippocampal cultures resulted in larger presynaptic Ca$^{2+}$ transients, as already described in previous studies (Glebov et al., 2017; Jeans et al., 2017). In the present study, elevated presynaptic Ca$^{2+}$ transients were exclusively observed in more mature cultures (18–22 DIV) following incubation with TTX and stimulation with 1 or 10 APs (Fig. 3). In less mature cultures at DIV 14−15, TTX did not induce a statistically significant increase in presynaptic Ca$^{2+}$ flux, although data points were highly variable. This is in line with previous findings of HSP adaptations being mostly postsynaptic in younger neurons, whereas presynaptic adaptations emerge as neurons mature (Han & Stevens, 2009; Wierenga et al., 2006). It would be interesting to corroborate findings from live cell imaging at DIV 14−15 with techniques such as western blotting and immunocytochemistry.

Presynaptic HSP involves dynamic restructuring of the active zone matrix (Lazarevic et al., 2011) and a recruitment of proteins of the presynaptic machinery (Glebov et al., 2017), including Ca$_V$2.1 channels (Glebov et al., 2017; Jeans et al., 2017). Our findings indicate

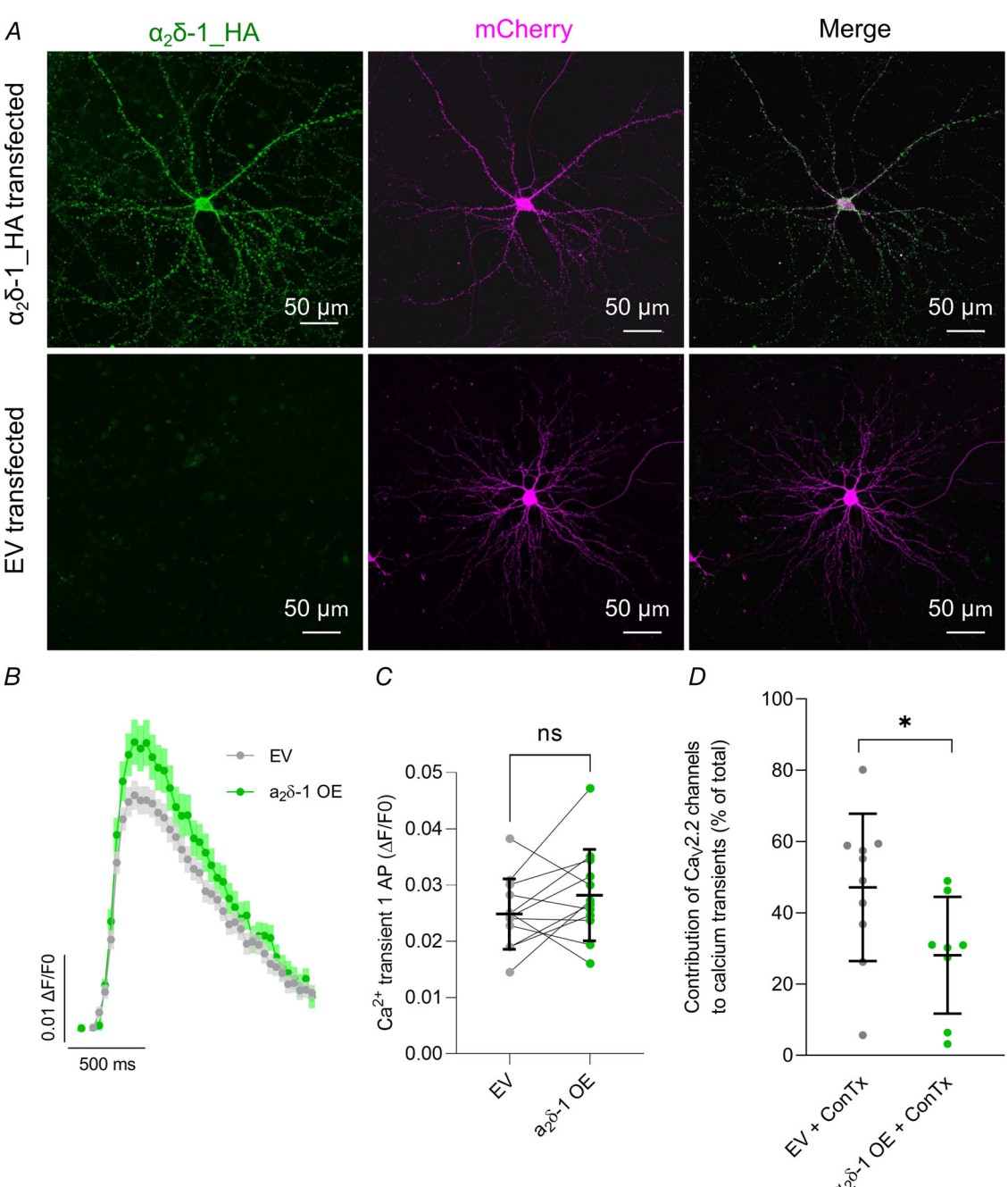

**Figure 5. Overexpression of $\alpha_2\delta$-1 does not change Ca$^{2+}$ transients resulting from one AP but decreases the contribution of Ca$_V$2.2 channels**

*A*, confocal images of immunostaining for $\alpha_2\delta$-1_HA in $\alpha_2\delta$-1-overexpressing neurons (top row) and control, empty vector (EV)-transfected neurons (bottom row). $\alpha_2\delta$-1_HA is shown in green and transfection marker mCherry in magenta. Maximum intensity projection of z-stacks and tile scan, optical section, 0.279 μm; confocal mode ×40; scale bar, 50 μm. *B*, similar Sy-GCaMP6f fluorescence changes after stimulation with one AP of EV (grey traces) and $\alpha_2\delta$-1-overexpressing (OE; green traces) neurons shown as means ± SEM. Averaged fluorescence traces of 5−8 repeats of stimulation with one AP; for each field of view, 10−75 ROIs were chosen for analysis of change in fluorescence over baseline fluorescence ($\Delta F/F_0$). *C*, Ca$^{2+}$ transient amplitudes after stimulation with one AP were similar in EV (grey) and $\alpha_2\delta$-1-overexpressing (OE; green) neurons. $n = 12$ biological replicates; paired $t$ test, $P = 0.18$; $n$ corresponds to independent experiments and data are also shown as means ± SD (black). *D*, contribution of Ca$_V$2.2 channels to Ca$^{2+}$ transients after stimulation with one AP was 47.1 ± 20.65% (grey) in EV neurons and 28.1 ± 16.38% (green) in neurons overexpressing $\alpha_2\delta$-1. $n$ for EV control = 10 fields of view, $n$ for $\alpha_2\delta$-1 = 8 fields of view; $n$ corresponds to fields of view from five independent experiments and data are shown as means ± SD; two-tailed unpaired $t$ test, $P = 0.05$. [Colour figure can be viewed at wileyonlinelibrary.com]

that an increased contribution of Ca$_V$2.2 channels to pre-synaptic Ca$^{2+}$ flux (from about 30% to 50%, Fig. 4*B*, *D* and *G*) represents another component of presynaptic HSP restructuring in cultured hippocampal neurons. This is further confirmed by both western blotting and super-resolution microscopy of Ca$_V$2.2_HA$^{KI/KI}$ neurons, revealing increased levels of Ca$_V$2.2 channels following HSP induction of approximately 40%. The increase in Ca$_V$2.2 channel expression during HSP has not been previously described, though a recent study detected an enrichment of Cav2.2 channels at the active zone after TTX treatment using stochastic optical reconstruction microscopy (STORM) super-resolution imaging (Glebov et al., 2017). The precise details of pre-synaptic restructuring during HSP, as well as the relevance

of the change in relative composition of active zone Ca$_V$2 channels, remain to be fully deciphered.

## $\alpha_2\delta$-1 overexpression downregulates Ca$_V$2.2 channel involvement in Ca$^{2+}$ response to one AP stimulation and prevents HSP

$\alpha_2\delta$-1 subunits are important parameters in synaptic transmission, by regulating Ca$_V$2 channel abundance at the presynaptic membrane (Hoppa et al., 2012). In addition, $\alpha_2\delta$-1 potentially interacts with other proteins to modulate synaptic activity, independent from Ca$_V$2 channels (Dolphin, 2013; Schöpf et al., 2021). Here, we show that $\alpha_2\delta$-1 overexpression reduces the contribution

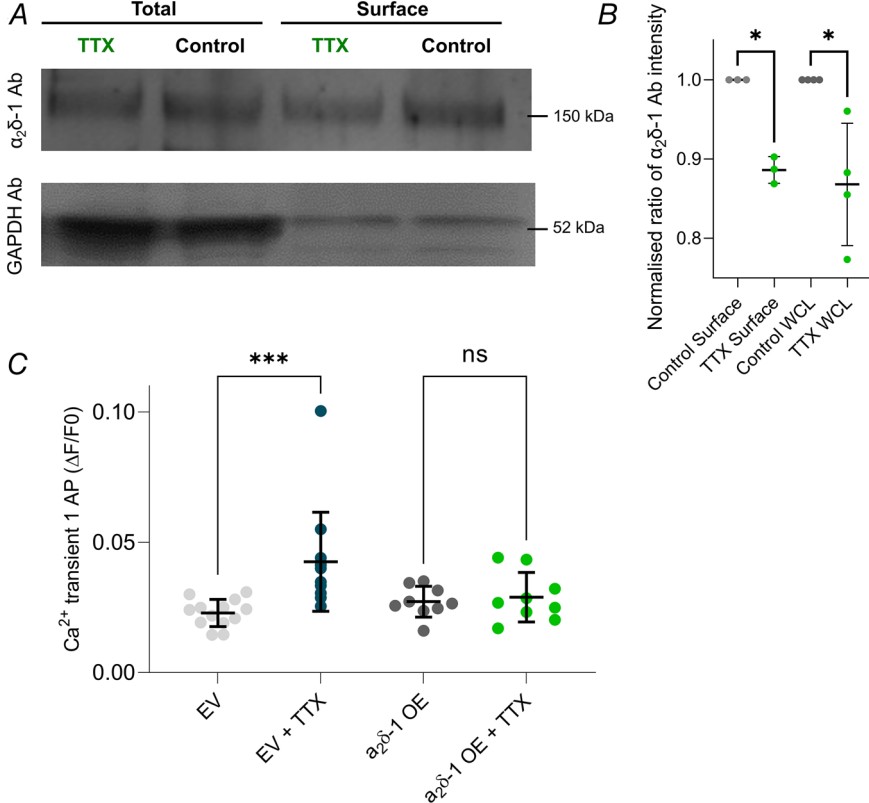

**Figure 6. During HSP, surface and total levels of endogenous $\alpha_2\delta$-1 decrease while overexpression of $\alpha_2\delta$-1 prevents HSP**

*A*, biotinylation experiments of control and TTX-treated hippocampal neurons showing the total endogenous amount of $\alpha_2\delta$-1 (WCL) and surface fractions, respectively (top gel). Bands for GAPDH are shown in the bottom gel. *B*, quantification of the intensity of endogenous $\alpha_2\delta$-1 protein bands reveals decreased levels of total and surface $\alpha_2\delta$-1 after the induction of HSP with TTX (green data points) compared to control, untreated neurons (grey data points). Data have been normalised to controls in each experiment, and analysed by one-sample two-tailed *t* test compared to a theoretical value of 1. WCL *P* = 0.0072 and surface *P* = 0.042; *n* for WCL = 4 independent experiments, *n* for surface fraction = 3 independent experiments, averaged duplicates for each experiment. *C*, Ca$^{2+}$ transient amplitudes after stimulation with one AP are larger in TTX-treated EV than in control EV conditions (grey data points) but similar in $\alpha_2\delta$-1-overexpressing neurons with (green) and without TTX (dark grey data points). One-way ANOVA, *F*(3,40) = 6.484, *P* = 0.001, Tukey's multiple comparisons test, EV *vs.* EV + TTX, *P* = 0.0008; $\alpha_2\delta$-1 *vs.* $\alpha_2\delta$-1 + TTX, *P* = 0.99; *n* corresponds to independent experiments and data are shown as means ± SD; *n* for EV and EV + TTX = 13, *n* for $\alpha_2\delta$-1 and $\alpha_2\delta$-1 + TTX = 9. [Colour figure can be viewed at wileyonlinelibrary.com]

of $Ca_V2.2$ channels to total $Ca^{2+}$ flux without altering the amplitude of the $Ca^{2+}$ transients. In contrast, previous work showed that $\alpha_2\delta$-1 overexpression reduced $Ca^{2+}$ flux relative to synaptic vesicular release (Hoppa et al., 2012). One possible explanation for our findings is that in more mature cultures, the downregulation of $Ca_V2.2$ channel plasticity as a result of $\alpha_2\delta$-1 overexpression was accompanied by a compensatory upregulation of other $Ca_V2$ channels, ensuring stable $Ca^{2+}$ transients.

Generally, the amount of $\alpha_2\delta$ proteins is likely to vary in different synapses, but levels are usually thought to be higher than those of $Ca_V2$ channels (Muller et al., 2010). It was shown that the complex between $\alpha_2\delta$ and $Ca_V2$ channels is easily disrupted (Muller et al., 2010; Voigt et al., 2016), and hence the more $\alpha_2\delta$ is present, the more $Ca_V2$ channels will be in a complex with $\alpha_2\delta$. $\alpha_2\delta$ proteins are glycosyl-phosphatidylinositol-anchored (Davies et al., 2010) and are present in lipids rafts, which are small microdomains within the plasma membrane, high in cholesterol and sphingolipids (for review see Pani & Singh, 2009). Notably, $\alpha_2\delta$ proteins also mediate the partitioning of $Ca_V2.1$ and $Ca_V2.2$ channels into these specialised lipid-rich membrane domains (Davies et al., 2006; Robinson et al., 2010), resulting in reduced $Ca^{2+}$ currents (Davies et al., 2006; Ronzitti et al., 2014). Increasing the abundance of $\alpha_2\delta$-1 in presynaptic terminals by overexpression may lead to increased levels of $Ca_V2.2$ channels in lipid rafts, potentially clamping their mobility, and their contribution to the plasticity of $Ca^{2+}$ flux. The fact that $Ca^{2+}$ transients in response to one AP did not change in $\alpha_2\delta$-1-overexpressing terminals may be due to compensatory $Ca_V2.1$ or $Ca_V2.3$ channel upregulation, ensuring stable $Ca^{2+}$ transients. Further work is needed to evaluate the stability of the interaction between $\alpha_2\delta$-1 and $Ca_V2$ channels at the different compartments of the active zone.

Furthermore, $\alpha_2\delta$-1 overexpression prevents elevated presynaptic $Ca^{2+}$ transients observed after TTX treatment. A recent study investigated the effect of $\alpha_2\delta$-1 overexpression on the developmental of neuronal networks *in vitro* and discovered that neurons overexpressing $\alpha_2\delta$-1 exhibit spontaneous neuronal activity and increased presynaptic glutamate release (Bikbaev et al., 2020). This finding might indicate that neuronal network activity in $\alpha_2\delta$-1-overexpressing neurons was already increased, and therefore the application of TTX did not have a potentiating effect. This increased neuronal activity might also explain the downregulation of $Ca_V2.2$ channels. We therefore also sought to determine changes in endogenous surface $\alpha_2\delta$-1 during HSP by cell surface biotinylation and immunoblotting. This revealed lower levels of $\alpha_2\delta$-1 both in WCL, and specifically on the surface of neurons after the induction of HSP with TTX, indicating that a downregulation of $\alpha_2\delta$-1 may contribute to processes involved in HSP.

Chronic silencing of neuronal activity thus caused an increase in $Ca^{2+}$ transients due to greater $Ca_V2.2$ channel contribution and increased $Ca_V2.2$ protein levels, while levels of $\alpha_2\delta$-1 decreased. This decrease of $\alpha_2\delta$-1 may therefore be required to allow increased mobility of $Ca_V2.2$ channels, necessary for presynaptic potentiation. Overexpression of $\alpha_2\delta$-1 potentially prevents the elevation of presynaptic $Ca^{2+}$ transients after TTX treatment by 'clamping' $Ca_V2.2$ channels in microdomains within the plasma membrane. Further experiments are required to provide mechanistic insight into $Ca_V2.2$ and $\alpha_2\delta$-1proteins during active zone restructuring during HSP.

Together, these findings show an involvement of $Ca_V2.2$ channels in HSP and prompt further examination into the role of $\alpha_2\delta$ proteins, as major regulators of homeostatic processes at synapses.

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

## Additional information

### Data availability statement

All data supporting the results in the paper are uploaded as Supporting Information.

### Competing interests

None.

### Author contributions

Conceptualisation: K.S.P and A.C.D. Experiments and analysis: K.S.P and K.H.R. Writing: K.S.P. Funding acquisition: A.C.D. All authors commented on the paper. All authors have read and approved the final version of this manuscript and agree to be accountable for all aspects of the work in ensuring that questions related to the accuracy or integrity of any part of the work are appropriately investigated and resolved. All persons designated as authors qualify for authorship, and all those who qualify for authorship are listed.

### Funding

None.

### Acknowledgements

We thank Dr Laurent Ferron for his help with initial live cell imaging experiments and Dr Ivan Kadurin for his help with

initial western blotting experiments. We thank Wendy S. Pratt and Kanchan Chaggar for skilful technical assistance and Stuart Martin for genotyping. We thank Dr Joshua Elliott for help with proof-reading the manuscript.

## Keywords

$\alpha_2\delta$-1, Ca$_V$2.2 channels, calcium imaging, homeostatic synaptic plasticity

## Supporting information

Additional supporting information can be found online in the Supporting Information section at the end of the HTML view of the article. Supporting information files available:

**Statistical Summary Document**
**Peer Review History**
**Original western blots**
**Data for Figs. 1–6**

