## [Peer Review History · The Journal of Physiology]

Involvement of CaV2.2 channels and $\alpha 2\delta$ -1 in hippocampal homeostatic synaptic plasticity

Kjara S Pilch, Krishma H Ramgoolam, and Annette C. Dolphin
DOI: 10.1113/JP283600

Corresponding author(s): Annette Dolphin (a.dolphin@ucl.ac.uk)

The following individual(s) involved in review of this submission have agreed to reveal their identity: Rajesh Khanna (Referee #1)

Review Timeline:

Submission Date:	15-Jul-2022
Editorial Decision:	31-Aug-2022
Revision Received:	15-Sep-2022
Editorial Decision:	17-Oct-2022
Revision Received:	01-Nov-2022
Accepted:	08-Nov-2022

Senior Editor: Katalin Toth

Reviewing Editor: Samuel Young

Transaction Report:

Dear Professor Dolphin,

Re: JP-RP-2022-283600 "Involvement of CaV2.2 channels and $\alpha 2\delta$ -1 in hippocampal homeostatic synaptic plasticity" by Annette C. Dolphin, Kjara S Pilch, and Krishma H Ramgoolam

Thank you for submitting your manuscript to The Journal of Physiology. It has been assessed by a Reviewing Editor and by 2 expert Referees and I am pleased to tell you that it is considered to be acceptable for publication following satisfactory revision.

The reports are copied at the end of this email. Please address all of the points and incorporate all requested revisions, or explain in your Response to Referees why a change has not been made.

NEW POLICY: In order to improve the transparency of its peer review process The Journal of Physiology publishes online as supporting information the peer review history of all articles accepted for publication. Readers will have access to decision letters, including all Editors' comments and referee reports, for each version of the manuscript and any author responses to peer review comments. Referees can decide whether or not they wish to be named on the peer review history document.

Authors are asked to use The Journal's premium BioRender (<https://biorender.com/>) account to create/redraw their Abstract Figures. Information on how to access The Journal's premium BioRender account is here:

<https://physoc.onlinelibrary.wiley.com/journal/14697793/biorender-access> and authors are expected to use this service. This will enable Authors to download high-resolution versions of their figures. The link provided should only be used for the purposes of this submission. Authors will be charged for figures created on this premium BioRender account if they are not related to this manuscript submission.

I hope you will find the comments helpful and have no difficulty returning your revisions within 4 weeks.

Your revised manuscript should be submitted online using the links in Author Tasks: Link Not Available.

Any image files uploaded with the previous version are retained on the system. Please ensure you replace or remove all files that have been revised.

REVISION CHECKLIST:

- Article file, including any tables and figure legends, must be in an editable format (eg Word)
- Abstract figure file (see above)
- Statistical Summary Document
- Upload each figure as a separate high quality file
- Upload a full Response to Referees, including a response to any Senior and Reviewing Editor Comments;
- Upload a copy of the manuscript with the changes highlighted.

- A potential 'Cover Art' file for consideration as the Issue's cover image;
- Appropriate Supporting Information (Video, audio or data set https://jp.msubmit.net/cgi-bin/main.plex?form_type=display_requirements#supp).

To create your 'Response to Referees' copy all the reports, including any comments from the Senior and Reviewing Editors, into a Word, or similar, file and respond to each point in colour or CAPITALS and upload this when you submit your revision.

I look forward to receiving your revised submission.

If you have any queries please reply to this email and staff will be happy to assist.

Yours sincerely,

Katalin Toth
Senior Editor
The Journal of Physiology

REQUIRED ITEMS:

- Author photo and profile. First (or joint first) authors are asked to provide a short biography (no more than 100 words for one author or 150 words in total for joint first authors) and a portrait photograph. These should be uploaded and clearly labelled with the revised version of the manuscript. See Information for Authors for further details.
- The contact information provided for the person responsible for 'Research Governance' at your institution is an author on this paper. Please provide an alternative contact who is not an author on this paper or confirm that the author whose email was provided has sole responsibility for research governance. This is the person who is responsible for regulations, principles and standards of good practice in research carried out at the institution, for instance the ethical treatment of animals, the keeping of proper experimental records or the reporting of results.
- You must start the Methods section with a paragraph headed Ethical Approval. A detailed explanation of journal policy and regulations on animal experimentation is given in Principles and standards for reporting animal experiments in The Journal of Physiology and Experimental Physiology by David Grundy J Physiol, 593: 2547-2549. doi:10.1113/JP270818). A checklist outlining these requirements and detailing the information that must be provided in the paper can be found at: <https://physoc.onlinelibrary.wiley.com/hub/animal-experiments>. Authors should confirm in their Methods section that their experiments were carried out according to the guidelines laid down by their institution's animal welfare committee, and conform to the principles and regulations as described in the Editorial by Grundy (2015). The Methods section must contain details of the anaesthetic regime: anaesthetic used, dose and route of administration and method of killing the experimental animals.
- The Journal of Physiology funds authors of provisionally accepted papers to use the premium BioRender site to create high resolution schematic figures. Follow this link and enter your details and the manuscript number to create and download figures. Upload these as the figure files for your revised submission. If you choose not to take up this offer we require figures to be of similar quality and resolution. If you are opting out of this service to authors, state this in the Comments section on the Detailed Information page of the submission form. The link provided should only be used for the purposes of this submission. Authors will be charged for figures created on this premium BioRender account if they are not related to this manuscript submission.
- Please upload separate high-quality figure files via the submission form.
- Your paper contains Supporting Information of a type that we no longer publish. Any information essential to an understanding of the paper must be included as part of the main manuscript and figures. The only Supporting Information that we publish are video and audio, 3D structures, program codes and large data files. Your revised paper will be returned to you if it does not adhere to our Supporting Information Guidelines.
- A Statistical Summary Document, summarising the statistics presented in the manuscript, is required upon revision. It must be on the Journal's template, which can be downloaded from the link in the Statistical Summary Document section here: https://jp.msubmit.net/cgi-bin/main.plex?form_type=display_requirements#statistics.
- Papers must comply with the Statistics Policy: https://jp.msubmit.net/cgi-bin/main.plex?form_type=display_requirements#statistics.

In summary:

- If $n \leq 30$, all data points must be plotted in the figure in a way that reveals their range and distribution. A bar graph with data points overlaid, a box and whisker plot or a violin plot (preferably with data points included) are acceptable formats.
- If $n > 30$, then the entire raw dataset must be made available either as supporting information, or hosted on a not-for-profit repository e.g. FigShare, with access details provided in the manuscript.
- 'n' clearly defined (e.g. x cells from y slices in z animals) in the Methods. Authors should be mindful of pseudoreplication.
- All relevant 'n' values must be clearly stated in the main text, figures and tables, and the Statistical Summary Document (required upon revision).
- The most appropriate summary statistic (e.g. mean or median and standard deviation) must be used. Standard Error of the

Mean (SEM) alone is not permitted.

- Exact p values must be stated. Authors must not use 'greater than' or 'less than'. Exact p values must be stated to three significant figures even when 'no statistical significance' is claimed.

- Statistics Summary Document completed appropriately upon revision.

- A Data Availability Statement is required for all papers reporting original data. This must be in the Additional Information section of the manuscript itself. It must have the paragraph heading "Data Availability Statement". All data supporting the results in the paper must be either: in the paper itself; uploaded as Supporting Information for Online Publication; or archived in an appropriate public repository. The statement needs to describe the availability or the absence of shared data. Authors must include in their Statement: a link to the repository they have used, or a statement that it is available as Supporting Information; reference the data in the appropriate sections(s) of their manuscript; and cite the data they have shared in the References section. Whenever possible the scripts and other artefacts used to generate the analyses presented in the paper should also be publicly archived. If sharing data compromises ethical standards or legal requirements then authors are not expected to share it, but must note this in their Statement. For more information, see our Statistics Policy.

- Please include an Abstract Figure. The Abstract Figure is a piece of artwork designed to give readers an immediate understanding of the research and should summarise the main conclusions. If possible, the image should be easily 'readable' from left to right or top to bottom. It should show the physiological relevance of the manuscript so readers can assess the importance and content of its findings. Abstract Figures should not merely recapitulate other figures in the manuscript. Please try to keep the diagram as simple as possible and without superfluous information that may distract from the main conclusion(s). Abstract Figures must be provided by authors no later than the revised manuscript stage and should be uploaded as a separate file during online submission labelled as File Type 'Abstract Figure'. Please ensure that you include the figure legend in the main article file. All Abstract Figures should be created using BioRender. Authors should use The Journal's premium BioRender account to export high-resolution images. Details on how to use and access the premium account are included as part of this email.

EDITOR COMMENTS

Reviewing Editor:

This manuscript provides several novel findings about Cav2.2 and alpha-2-delta-1 contributions to homeostatic plasticity. Both reviewers felt that the findings were highly significant to the field, very interesting and addressed important questions. Despite the enthusiasm for the manuscript there are issues to be addressed. An important concern is the classification of young hippocampal neurons as DIV14-17 and mature hippocampal neurons as DIV18-22. Given that there is large variability in developmental characteristics in primary cultured neurons, a more restricted range of development time windows should be analyzed. The authors should perform analysis and compare cultures from DIV14-16 to cultures from DIV20 onwards or DIV14-17 to DIV21 onwards. This would allow for clear conclusions to be drawn about developmental changes. The authors only show data from single AP recordings, however in the methods they state they have also done 10 AP recordings. Since the authors have this data in hand, it would greatly strengthen conclusions and provide valuable data to the field to show the 10 AP data in the manuscript. All statistics must be reported as SD, not SEM. Please address concerns about statistics. Finally, the manuscript writing is a bit disjointed, and the authors should carefully revise and rewrite their manuscript to address this concern.

REFEREE COMMENTS

Referee #1:

Please see the attached file.

Referee #2:

The presented manuscript by Pilch et al. addresses important topics, namely the involvement of the largely presynaptic calcium channel Cav2.2 as well as the auxiliary channel subunit alpha-2-delta-1 in homeostatic plasticity. Using qPCR they first show the predominant expression of Cav2.2 transcripts in cortex and hippocampus of adult mice. Protein biochemistry on a previously developed knock-in mouse model (Cav2.2 channel with an HA tag) supports the findings on mRNA. Using a genetically encoded calcium sensor they demonstrate that the induction of homeostatic plasticity by a well-established experimental paradigm (48 h TTX application) increases presynaptic calcium transients in mature, but not in young neurons. Calcium signals after inducing homeostatic plasticity reveal an increased N-type contribution, which is likely due to increased protein expression, as shown by protein biochemistry of endogenous Cav2.2 and immunostaining of HA-coupled channels. In another set of experiments the authors show that overexpression of alpha-2-delta-1 does not alter the presynaptic calcium signal, however, it decreases the N-type channel contribution. Finally, they demonstrate that after inducing homeostatic plasticity the level of alpha-2-delta-1 decreases and that overexpression of alpha-2-delta-1 prevents the

plasticity-induced increase in synaptic calcium signals.

Based on this brief summary it is evident that the study addresses several important topics, which all may provide insights into general synapse physiology and the mechanisms involved in plasticity. Unfortunately, a number of points limit the impact of the present study:

1. The study presents several findings that are interesting by themselves but largely descriptive and not necessarily well connected with each other. The finding that Cav2.2 channel contribution increases after inducing homeostatic plasticity is interesting and novel. So are the findings that alpha-2-delta-1 levels are altered by homeostatic plasticity, that overexpression of alpha-2-delta-1 prevents homeostatic effects on synaptic calcium signals, and that overexpression of alpha-2-delta-1 decreases the N-type channel contribution to calcium signals. Each finding is interesting, raises important questions and the underlying mechanisms need to be discussed/addressed in future experiments. The findings do, however, not explain each other and therefore not provide substantial mechanistic insight.
2. The key findings of the study are based on the measurement of presynaptic calcium signals using the genetically encoded calcium sensor GCaMP6f and only take signals after stimulation with 1 AP into account. Measurements have been replicated to reduce the expected variability. Still, substantial variability is expected to underlie signals triggered by 1 AP. Stimulations with 10 APs, which generate more robust responses (compare figures 2 D and E), are not taken into account for the main conclusions.
3. The developmental windows selected for the cultured neurons (young DIV14-17, mature DIV18-22) are not sharply separating developmental phases (if DIV 14-17 is selected for young cultures, mature cultures should be analyzed after DIV 21). This developmental distinction is, however, important for the main conclusions of the manuscript.
4. Statistical interpretation partly under- and partly overestimates changes. For example, similar trends are observed in younger neurons, yet the differences are not statistically significant and hence interpreted as "no effect". Similarly, overexpression of direction of alpha-2-delta-1 slightly, but not significantly, increases calcium signals. These experiments would benefit strongly from increasing n-numbers (more neurons analyzed from the same cultures instead of replicated measurements of the same selected regions - see also examples below) and, as mentioned above, from also taking responses with more robust stimulation paradigms into account. Regarding statistical analyses, see also specific points below.

Specific points:

1. Title: "Involvement of CaV2.2 channels and $\alpha 2\delta$ -1 in hippocampal homeostatic synaptic plasticity" should include "culture hippocampal neurons" rather than "hippocampal" (hippocampus was not studied).
2. Running title: "CaV2.2 and $\alpha 2\delta$ -1 in homeostatic plasticity" is more appropriate.
3. Introduction: [...] Here, we show that CaV2.2 channels are involved in HSP in the hippocampus and [...]: "in hippocampal neurons" instead of "in the hippocampus" (again, plasticity in the hippocampus was not studied in the present manuscript).
4. Even though I understand the intention (mouse model) the term "endogenous CaV2.2_HA" is a contradiction in itself, please avoid or be more precise here.
5. Clearly define the developmental stages (e.g., "adult") also in the main text, not only in methods.
6. Fig. 2A-C: what are the arrowheads pointing at (exemplary boutons?)
7. Fig. 2D-E: precise n-numbers are missing.
8. Discussion, first paragraph: "the role of CaV2.2 channels for synaptic transmission in the hippocampus" was not studied, so weren't "hippocampal synapses".
9. Discussion: "...our findings indicate that this developmental down-regulation of CaV2.2 channels may not relate to the majority of cortical and hippocampal synapses." This is an invalid conclusion. Relative Cav2.2 levels were studied but not compared with Cav2.1 levels in the present study. The relative distribution of Cav2.2 channels does not allow conclusions on the distribution of Cav2.1 channels.

Statistics:

1. terms like F(1.1, 2.2) or F(1.6, 4.9) are not possible, please check and correct throughout the manuscript.
2. I agree with the authors that in some experiments the results need to be normalized to the respective controls. But it needs to be mentioned when this was done, and the variability of the control group needs to be maintained. Otherwise, the selected analyses are not valid. (Fig. 1G-I, Fig. 4D, and 6B).
3. Provide precise information throughout on: What are biological replicates (cultures or cells)? How many cells have been

analyzed from each culture preparation? Why are the data analyzed and presented as paired? Example figure 3: "boutons, n = 8 biological replicates, two-tailed paired t-test P = 0.16; t = 1.56; df = 7; n corresponds to independent experiments and [...]" Example figure 5: "n for EV control = 10 fields of view and n for $\alpha 2\delta$ -1 = 8 fields of view. n corresponds to fields of view from 5 independent experiments" does this mean that for some culture preparations only 1 field of view was analyzed?

4. Fig. 5C: The mean plus/minus SEM bars do not at all match the displayed data points!

5. Toxin data (Fig. 4 and 5) should be supplemented with mean traces of calcium signals (before and after application). Why are these data (Fig. 4, before and after ConTx application) not presented and analyzed as "paired"? Or was the ConTx treatment performed on different cells?

6. Fig. 6C: If a one-way ANOVA was performed the data cannot be presented as paired.

Editorials:

1. Check spaces before citations throughout.

2. p.11, last sentence: "anti- $\alpha 2\delta$ -1" should read "anti-HA".

3. check X-axis labelling of figure 6C.

4. Discussion: "Although the developmental shift from CaV2.1 to CaV2.2 channels" guess the authors mean from Cav2.2 to Cav2.1?

5. Methods: "with 1.5 mM tetrodotoxin" Guess the authors mean micromolar?

END OF COMMENTS

Confidential Review

15-Jul-2022

In this manuscript, the Dolphin lab provides evidence for a novel role for Cav2.2 in synaptic transmission by demonstrating a direct relationship between increased Cav2.2 and calcium influx during high frequency stimulation. They also go on to demonstrate high frequency stimulation decreases levels of the auxiliary subunit $\alpha_2\delta$ while an increase in $\alpha_2\delta$ normalizes the Cav2.2 effect on presynaptic calcium influx. Some of the data on Cav2.2 is largely confirmatory – albeit in a different synapse. Overall, this is an interesting study with excellent rigor, high quality data, superb writing that does not overinterpret data. One minor concern is why the authors did not evaluate the contribution of Cav2.1 and Cav2.3 in the experiments. Notwithstanding this minor limitation, overall, this is an interesting study with excellent rigor, high quality data, appropriate statistical analyses, and superb writing that does not overinterpret data.

Response to Editor and Reviewers

REVIEWING EDITOR:

This manuscript provides several novel findings about Cav2.2 and alpha-2-delta-1 contributions to homeostatic plasticity. Both reviewers felt that the findings were highly significant to the field, very interesting and addressed important questions. Despite the enthusiasm for the manuscript there are issues to be addressed. An important concern is the classification of young hippocampal neurons as DIV14-17 and mature hippocampal neurons as DIV18-22. Given that there is large variability in developmental characteristics in primary cultured neurons, a more restricted range of development time windows should be analyzed. The authors should perform analysis and compare cultures from DIV14-16 to cultures from DIV20 onwards or DIV14-17 to DIV21 onwards. This would allow for clear conclusions to be drawn about developmental changes. The authors only show data from single AP recordings, however in the methods they state they have also done 10 AP recordings. Since the authors have this data in hand, it would greatly strengthen conclusions and provide valuable data to the field to show the 10 AP data in the manuscript. All statistics must be reported as SD, not SEM. Please address concerns about statistics. Finally, the manuscript writing is a bit disjointed, and the authors should carefully revise and rewrite their manuscript to address this concern.

Response: See response to reviewers, we have added 10 AP data and SD instead of SEM for all graphs. Additionally, we have included in Fig 1 parts K-L from Supplementary data.

REFEREE COMMENTS

Referee #1:

In this manuscript, the Dolphin lab provides evidence for a novel role for Cav2.2 in synaptic transmission by demonstrating a direct relationship between increased Cav2.2 and calcium influx during high frequency stimulation. They also go on to demonstrate high frequency stimulation decreases levels of the auxiliary subunit a2d while an increase in a2d normalizes the Cav2.2 effect on presynaptic calcium influx. Some of the data on Cav2.2 is largely confirmatory – albeit in a different synapse. Overall, this is an interesting study with excellent rigor, high quality data, superb writing that does not overinterpret data. One minor concern is why the authors did not evaluate the contribution of Cav2.1 and Cav2.3 in the experiments. Notwithstanding this minor limitation, overall, this is an interesting study with excellent rigor, high quality data, appropriate statistical analyses, and superb writing that does not overinterpret data.

Response: We thank the reviewer for their comments. It would have been very interesting to be able to compare the contribution of Ca_v2.2 channels to Ca_v2.1 and Ca_v2.3, however performing these experiments was out of the scope of this work. Some experiments apply ConTx, Agatoxin and SXN consecutively on live neurons during calcium imaging to dissect the contribution of each Ca_v2 type. However, I would be concerned about an artificial reduction of GCaMP6f fluorescence as cells bleach and show reduced activity after repetitive stimulation and waiting time. Moreover, visualising endogenous Ca_v2 channels in culture is challenging; for this reason, we developed the Ca_v2.2 KI

mouse allowing us to track this specific channel. Further experiments should aim to provide a comparative overview of all three channels during different types of synaptic plasticity.

Referee #2:

The presented manuscript by Pilch et al. addresses important topics, namely the involvement of the largely presynaptic calcium channel Cav2.2 as well as the auxiliary channel subunit alpha-2-delta-1 in homeostatic plasticity. Using qPCR they first show the predominant expression of Cav2.2 transcripts in cortex and hippocampus of adult mice. Protein biochemistry on a previously developed knock-in mouse model (Cav2.2 channel with an HA tag) supports the findings on mRNA. Using a genetically encoded calcium sensor they demonstrate that the induction of homeostatic plasticity by a well-established experimental paradigm (48 h TTX application) increases presynaptic calcium transients in mature, but not in young neurons. Calcium signals after inducing homeostatic plasticity reveal an increased N-type contribution, which is likely due to increased protein expression, as shown by protein biochemistry of endogenous Cav2.2 and immunostaining of HA-coupled channels. In another set of experiments the authors show that overexpression of alpha-2-delta-1 does not alter the presynaptic calcium signal, however, it decreases the N-type channel contribution. Finally, they demonstrate that after inducing homeostatic plasticity the level of alpha-2-delta-1 decreases and that overexpression of alpha-2-delta-1 prevents the plasticity-induced increase in synaptic calcium signals.

Based on this brief summary it is evident that the study addresses several important topics, which all may provide insights into general synapse physiology and the mechanisms involved in plasticity.

Unfortunately, a number of points limit the impact of the present study:

1. The study presents several findings that are interesting by themselves but largely descriptive and not necessarily well connected with each other. The finding that Cav2.2 channel contribution increases after inducing homeostatic plasticity is interesting and novel. So are the findings that alpha-2-delta-1 levels are altered by homeostatic plasticity, that overexpression of alpha-2-delta-1 prevents homeostatic effects on synaptic calcium signals, and that overexpression of alpha-2-delta-1 decreases the N-type channel contribution to calcium signals. Each finding is interesting, raises important questions and the underlying mechanisms need to be discussed/addressed in future experiments. The findings do, however, not explain each other and therefore not provide substantial mechanistic insight.

Response: In the Discussion we have tried to bring the results together with speculation about the role of $\alpha_2\delta$ -1; future studies will be needed to examine this in more detail, and whether other $\alpha_2\delta$ proteins may also be involved.

2. The key findings of the study are based on the measurement of presynaptic calcium signals using the genetically encoded calcium sensor GCaMP6f and only take signals after stimulation with 1 AP into account. Measurements have been replicated to reduce the expected variability. Still, substantial variability is expected to underlie signals triggered by 1 AP. Stimulations with 10 APs, which generate

more robust responses (compare figures 2 D and E), are not taken into account for the main conclusions.

Response: We thank the reviewer for the suggestions and added the data obtained with 10 AP stimulations that show a similar trend to the 1 AP stimulations (Fig 3).

3. The developmental windows selected for the cultured neurons (young DIV14-17, mature DIV18-22) are not sharply separating developmental phases (if DIV 14-17 is selected for young cultures, mature cultures should be analyzed after DIV 21). This developmental distinction is, however, important for the main conclusions of the manuscript.

Response: This is an excellent suggestion, providing a clearer distinction between the two ages. We have removed data points from DIV 16 and 17 and now have a sharper distinction of less mature neurons at DIV 14-15 and more mature neurons at DIV 18-22.

4. Statistical interpretation partly under- and partly overestimates changes. For example, similar trends are observed in younger neurons, yet the differences are not statistically significant and hence interpreted as "no effect". Similarly, overexpression of direction of alpha-2-delta-1 slightly, but not significantly, increases calcium signals. These experiments would benefit strongly from increasing n-numbers (more neurons analyzed from the same cultures instead of replicated measurements of the same selected regions - see also examples below) and, as mentioned above, from also taking responses with more robust stimulation paradigms into account. Regarding statistical analyses, see also specific points below.

Response: We thank the reviewer for these comments. We are unfortunately not in a position to increase the n numbers for these experiments, particularly given the time window for the minor revision of the manuscript that is requested, and the limited time that Dr. Pilch has before starting a new position. However, we believe that it is possible to draw some conclusions from the current data. We agree that it would be interesting to confirm findings from calcium imaging experiments in younger cultures DIV 14-15 using different techniques including western blotting. These experiments should be performed in the future and we have added a sentence to the discussion accordingly.

Specific points:

1. Title: "Involvement of CaV2.2 channels and $\alpha 2\delta$ -1 in hippocampal homeostatic synaptic plasticity" should include "culture hippocampal neurons" rather than "hippocampal" (hippocampus was not studied).

Response: Corrected.

2. Running title: "CaV2.2 and $\alpha 2\delta$ -1 in homeostatic plasticity" is more appropriate.

Response: Corrected.

3. Introduction: [...] Here, we show that CaV2.2 channels are involved in HSP in the hippocampus and

[...]: "in hippocampal neurons" instead of "in the hippocampus" (again, plasticity in the hippocampus was not studied in the present manuscript).

Response: Corrected.

4. Even though I understand the intention (mouse model) the term "endogenous CaV2.2_HA" is a contradiction in itself, please avoid or be more precise here.

Response: Corrected to endogenous Cav2.2.

5. Clearly define the developmental stages (e.g., "adult") also in the main text, not only in methods.

Response: Details added.

6. Fig. 2A-C: what are the arrowheads pointing at (exemplary boutons?)

Response: Details added.

7. Fig. 2D-E: precise n-numbers are missing.

Response: Added.

8. Discussion, first paragraph: "the role of CaV2.2 channels for synaptic transmission in the hippocampus" was not studied, so weren't "hippocampal synapses".

Response: Corrected.

9. Discussion: "...our findings indicate that this developmental down-regulation of CaV2.2 channels may not relate to the majority of cortical and hippocampal synapses." This is an invalid conclusion. Relative Cav2.2 levels were studied but not compared with Cav2.1 levels in the present study. The relative distribution of Cav2.2 channels does not allow conclusions on the distribution of Cav2.1 channels.

Response: Relative Cav2.2 levels were studied but not compared with Cav2.1 levels in the present study. We agree, the relative distribution of Cav2.2 channels does not allow conclusions on the distribution of Cav2.1 channels. We have changed this sentence in the Discussion.

Statistics:

1. terms like $F(1.1, 2.2)$ or $F(1.6, 4.9)$ are not possible, please check and correct throughout the manuscript.

Response: Corrected.

2. I agree with the authors that in some experiments the results need to be normalized to the respective controls. But it needs to be mentioned when this was done, and the variability of the control group needs to be maintained. Otherwise, the selected analyses are not valid. (Fig. 1G-I, Fig. 4D, and 6B).

Response: We thank the reviewer for their input on western blot analysis and realise this was not explained sufficiently in the Materials and Methods section. We have now explained the normalisation procedure in the Methods. In Figures 1G-I; 4F and 6B we cannot show the variability of the control group as we only have one value per experiment. Because absolute values of intensity of the blots are different between experiments, due to the technical issue that they are acutely dependent on western blot development and exposure times resulting in a variable background, therefore we cannot provide meaningful information on variability of controls between experiments, but only examine differences within experiments, and then compare between experiments following normalisation. This is different from the QPCR data where absolute values allow comparison between the controls.

3. Provide precise information throughout on: What are biological replicates (cultures or cells)? How many cells have been analyzed from each culture preparation? Why are the data analyzed and presented as paired? Example figure 3: "boutons, n = 8 biological replicates, two-tailed paired t-test P = 0.16; t = 1.56; df = 7; n corresponds to independent experiments and [...]" Example figure 5: "n for EV control = 10 fields of view and n for $\alpha 2\delta$ -1 = 8 fields of view. n corresponds to fields of view from 5 independent experiments" does this mean that for some culture preparations only 1 field of view was analyzed?

Response: We thank the reviewer for their comment. We have added details on the biological replicates and ensured that all figure legends contain sufficient information on sample size. For the live cell imaging experiments, sometimes only 1 field of view was analyzed per experiment. For example, when we applied the irreversible inhibitor Conotoxin to dishes, we would apply it to all cells of that dish simultaneously which is why we would then discard that dish. In addition, we would discard a dish if we had imaged it for one hour to avoid experimental artefacts from cells bleaching or responding less due to excessive electrical stimulation. In some conditions, we had only managed to fully image one field of view in that time (8 x 1 AP with 30 sec break in between, 2 min wait, 10 AP, 2 min wait, VAMPmOr2 stimulation 1 min, 5 min wait afterwards). This is explained in the live cell imaging section of Methods.

4. Fig. 5C: The mean plus/minus SEM bars do not at all match the displayed data points!

Response: Corrected.

5. Toxin data (Fig. 4 and 5) should be supplemented with mean traces of calcium signals (before and after application). Why are these data (Fig. 4, before and after ConTx application) not presented and analyzed as "paired"? Or was the ConTx treatment performed on different cells?

Response: We have now added the traces of pre-toxin and post toxin application to show the decrease in Sy-GCaMP6f fluorescence as $Ca_v2.2$ channels are blocked (Fig 4 A and C). The ConTx application was performed on different cells (either from control dishes or from dishes treated with TTX) which is why the analysis was not done as paired (not a before-after situation).

6. Fig. 6C: If a one-way ANOVA was performed the data cannot be presented as paired.

Response: Corrected.

Editorials:

1. Check spaces before citations throughout.

Response: Done.

2. p.11, last sentence: "anti- α 2 δ -1" should read "anti-HA".

Response: Corrected

3. check X-axis labelling of figure 6C.

Response: Corrected

4. Discussion: "Although the developmental shift from CaV2.1 to CaV2.2 channels" guess the authors mean from Cav2.2 to Cav2.1?

Response: Corrected.

5. Methods: "with 1.5 mM tetrodotoxin" Guess the authors mean micromolar?

Response: Corrected

Hi Annette,

Please note one additional requirement to your submission:

- You must upload original, uncropped western blot/gel images (including controls) if they are not included in the manuscript. This is to confirm that no inappropriate, unethical or misleading image manipulation has occurred <https://physoc.onlinelibrary.wiley.com/hub/journal-policies#imagmanip> These should be uploaded as 'Supporting information for review process only'. Please label/highlight the original gels so that we can clearly see which sections/lanes have been used in the manuscript figures.

Many thanks,

Paroma

Dear Professor Dolphin,

Re: JP-RP-2022-283600R1 "Involvement of CaV2.2 channels and $\alpha 2\delta$ -1 in hippocampal homeostatic synaptic plasticity" by Annette C. Dolphin, Kjara S Pilch, and Krishma H Ramgoolam

Thank you for submitting your manuscript to The Journal of Physiology. It has been assessed by a Reviewing Editor and by 2 expert Referees and I am pleased to tell you that it is considered to be acceptable for publication following satisfactory revision.

The reports are copied at the end of this email. Please address all of the points and incorporate all requested revisions, or explain in your Response to Referees why a change has not been made.

NEW POLICY: In order to improve the transparency of its peer review process The Journal of Physiology publishes online as supporting information the peer review history of all articles accepted for publication. Readers will have access to decision letters, including all Editors' comments and referee reports, for each version of the manuscript and any author responses to peer review comments. Referees can decide whether or not they wish to be named on the peer review history document.

Authors are asked to use The Journal's premium BioRender (<https://biorender.com/>) account to create/redraw their Abstract Figures. Information on how to access The Journal's premium BioRender account is here:

<https://physoc.onlinelibrary.wiley.com/journal/14697793/biorender-access> and authors are expected to use this service. This will enable Authors to download high-resolution versions of their figures. The link provided should only be used for the purposes of this submission. Authors will be charged for figures created on this premium BioRender account if they are not related to this manuscript submission.

I hope you will find the comments helpful and have no difficulty returning your revisions within 4 weeks.

Your revised manuscript should be submitted online using the links in Author Tasks Link Not Available.

Any image files uploaded with the previous version are retained on the system. Please ensure you replace or remove all files that have been revised.

REVISION CHECKLIST:

- Article file, including any tables and figure legends, must be in an editable format (eg Word)
- Abstract figure file (see above)
- Statistical Summary Document
- Upload each figure as a separate high quality file
- Upload a full Response to Referees, including a response to any Senior and Reviewing Editor Comments;

- Upload a copy of the manuscript with the changes highlighted.

- A potential 'Cover Art' file for consideration as the Issue's cover image;

- Appropriate Supporting Information (Video, audio or data set https://jp.msubmit.net/cgi-bin/main.plex?form_type=display_requirements#supp).

To create your 'Response to Referees' copy all the reports, including any comments from the Senior and Reviewing Editors, into a Word, or similar, file and respond to each point in colour or CAPITALS and upload this when you submit your revision.

I look forward to receiving your revised submission.

If you have any queries please reply to this email and staff will be happy to assist.

Yours sincerely,

Katalin Toth
Senior Editor
The Journal of Physiology

REQUIRED ITEMS:

-The contact information provided for the person responsible for 'Research Governance' at your institution is an author on this paper. Please provide an alternative contact who is not an author on this paper or confirm that the author whose email was provided has sole responsibility for research governance. This is the person who is responsible for regulations, principles and standards of good practice in research carried out at the institution, for instance the ethical treatment of animals, the keeping of proper experimental records or the reporting of results.

-You must start the Methods section with a paragraph headed Ethical Approval. A detailed explanation of journal policy and regulations on animal experimentation is given in Principles and standards for reporting animal experiments in The Journal of Physiology and Experimental Physiology by David Grundy (J Physiol, 593: 2547-2549. doi:10.1113/JP270818.). A checklist outlining these requirements and detailing the information that must be provided in the paper can be found at: <https://physoc.onlinelibrary.wiley.com/hub/animal-experiments>. Authors should confirm in their Methods section that their experiments were carried out according to the guidelines laid down by their institution's animal welfare committee, and conform to the principles and regulations as described in the Editorial by Grundy (2015). The Methods section must contain details of the anaesthetic regime: anaesthetic used, dose and route of administration and method of killing the experimental animals.

-Papers must comply with the Statistics Policy https://jp.msubmit.net/cgi-bin/main.plex?form_type=display_requirements#statistics

In summary:

-If $n \leq 30$, all data points must be plotted in the figure in a way that reveals their range and distribution. A bar graph with data points overlaid, a box and whisker plot or a violin plot (preferably with data points included) are acceptable formats.

-If $n > 30$, then the entire raw dataset must be made available either as supporting information, or hosted on a not-for-profit repository e.g. FigShare, with access details provided in the manuscript.

- n clearly defined (e.g. x cells from y slices in z animals) in the Methods. Authors should be mindful of pseudoreplication.

-All relevant n values must be clearly stated in the main text, figures and tables, and the Statistical Summary Document (required upon revision)

-The most appropriate summary statistic (e.g. mean or median and standard deviation) must be used. Standard Error of the

Mean (SEM) alone is not permitted.

-Exact p values must be stated. Authors must not use 'greater than' or 'less than'. Exact p values must be stated to three significant figures even when 'no statistical significance' is claimed.

-Statistics Summary Document completed appropriately upon revision

EDITOR COMMENTS

Reviewing Editor:

REFEREE COMMENTS

END OF COMMENTS

1st Confidential Review

15-Sep-2022

Response to Editor and Reviewers-----

EDITOR COMMENTS

Reviewing Editor:

The authors have done an excellent job of responding to the previous comments and concerns. However, there appear to still be concerns that need to be addressed. In particular, the authors should address why a data point was dropped and concern about proper use of statistics with the normalized data. Finally please explicitly state the method of animal euthanasia in methods section.

Response: 1) There was an error in the original data-set which has been corrected, as explained below. 2) We have now used a different statistical test, one sample t test, with the same outcomes. 3) We have added further information on animal euthanasia to the Methods.

REFEREE COMMENTS

Referee #1:

The authors have satisfactorily addressed my concerns.

Referee #2:

The authors have addressed all points raised by providing additional analyses, corrected analyses, as well as changes to the text. Particularly the additional analyses strengthen the main conclusions of the study. However, there are a few points from the initial discussion which require further clarification by the authors. Also, two new points arose due to changes incorporated into the revised manuscript:

Original major point 3: Authors response: This is an excellent suggestion, providing a clearer distinction between the two ages. We have removed data points from DIV 16 and 17 and now have a sharper distinction of less mature neurons at DIV 14-15 and more mature neurons at DIV 18-22.

Comment: The authors have addressed this point by apparently removing a data point from the original DIV 14-17 window reducing the n number from 8 to 7, which is acceptable. On the other hand, why was also one data point removed from the DIV 18-22 group, even though this time window was not altered?

Response: We thank the reviewer for the question. In re-checking all the data with respect to DIV, we realised there was a duplicate value in the data sheet which was removed for the revised version of the manuscript. We apologize for any confusion this may have caused.

Original specific point 4:

"endogenous" is also not correct, replace simply with "to visualize Cav2.2_HA"

Response: We have changed this

Statistics:

ad original point 1) Apparently there is a misunderstanding as the statistical information was not corrected but simply removed. Please provide the correct information (F value with correct information on degrees of freedom, p value of ANOVA) for all ANOVAs performed and presented in the manuscript. My initial point related to the fact that the degrees of freedom were represented by decimal numbers (original manuscript: F(1.1, 2.2) or F(1.6, 4.9)) which is not possible.

Response: We have added this information in the text and in the Statistics table.

ad original point 2) Ok regarding the explanation of the normalization procedure. However, in such cases (no variability in control values) t-test or ANOVA are not applicable and non-parametric statistical procedures have to be applied.

Response: We have now used a one sample t test compared to a theoretical mean of 1 for all normalised data. The normalised data are continuous and can both increase and decrease around 1.

New point 1: The Y-axis scaling of the figures 4F and 6C was change in the revised manuscript. However, figures should be displayed using the original Y-axis scaling.

Response: Fig. 4F, we changed labelling from "II-III loop intensity (normalised to GAPDH)" to "normalised ratio of II-III loop Ab intensity" as the latter is more correct.
Fig. 6C was not changed; Fig. 6B was changed from " $\alpha_2\delta$ -1 intensity" to "normalised ratio of $\alpha_2\delta$ -1 Ab intensity" as this is more correct.

New point 2: Why did the authors replace all SEM error bars with SD error bars throughout the revised manuscript? SD is irrelevant when comparing the probability of means, hence only SEM error bars must be shown.

Response: This is a requirement of the journal, although we agree with the reviewer.

Dear Dr Dolphin,

Re: JP-RP-2022-283600R2 "Involvement of CaV2.2 channels and $\alpha\delta$ -1 in hippocampal homeostatic synaptic plasticity" by Kjara S Pilch, Krishma H Ramgoolam, and Annette C. Dolphin

We are pleased to tell you that your paper has been accepted for publication in The Journal of Physiology.

Authors should note that it is too late at this point to offer corrections prior to proofing. The accepted version will be published online, ahead of the copy edited and typeset version being made available. Major corrections at proof stage, such as changes to figures, will be referred to the Editors for approval before they can be incorporated. Only minor changes, such as to style and consistency, should be made at proof stage. Changes that need to be made after proof stage will usually require a formal correction notice.

Yours sincerely,

Katalin Toth
Senior Editor
The Journal of Physiology

P.S. - You can help your research get the attention it deserves! Check out Wiley's free Promotion Guide for best-practice recommendations for promoting your work at www.wileyauthors.com/eoo/guide. You can learn more about Wiley Editing Services which offers professional video, design, and writing services to create shareable video abstracts, infographics, conference posters, lay summaries, and research news stories for your research at www.wileyauthors.com/eoo/promotion.

IMPORTANT NOTICE ABOUT OPEN ACCESS: To assist authors whose funding agencies mandate public access to published research findings sooner than 12 months after publication The Journal of Physiology allows authors to pay an Open Access (OA) fee to have their papers made freely available immediately on publication.

You can check if your funder or institution has a Wiley Open Access Account here: <https://authorservices.wiley.com/author-resources/Journal-Authors/licensing-and-open-access/open-access/author-compliance-tool.html>.

EDITOR COMMENTS

Reviewing Editor:

The authors have satisfied all comments and no further concerns exist.